Corrected: Publisher correction

# An expression atlas of variant ionotropic glutamate receptors identifies a molecular basis of carbonation sensing

Juan Antonio Sánchez-Alcañiz[1], Ana Florencia Silbering[1], Vincent Croset[1,5], Giovanna Zappia[1], Anantha Krishna Sivasubramaniam[1], Liliane Abuin[1], Saumya Yashmohini Sahai[2], Daniel Münch[3], Kathrin Steck[3], Thomas O. Auer[1], Steeve Cruchet[1], G. Larisa Neagu-Maier[4], Simon G. Sprecher [4], Carlos Ribeiro [3], Nilay Yapici [2] & Richard Benton [1]

Through analysis of the *Drosophila* ionotropic receptors (IRs), a family of variant ionotropic glutamate receptors, we reveal that most IRs are expressed in peripheral neuron populations in diverse gustatory organs in larvae and adults. We characterise IR56d, which defines two anatomically-distinct neuron classes in the proboscis: one responds to carbonated solutions and fatty acids while the other represents a subset of sugar- and fatty acid-sensing cells. Mutational analysis indicates that IR56d, together with the broadly-expressed co-receptors IR25a and IR76b, is essential for physiological responses to carbonation and fatty acids, but not sugars. We further demonstrate that carbonation and fatty acids both promote IR56d-dependent attraction of flies, but through different behavioural outputs. Our work provides a toolkit for investigating taste functions of IRs, defines a subset of these receptors required for carbonation sensing, and illustrates how the gustatory system uses combinatorial expression of sensory molecules in distinct neurons to coordinate behaviour.

[1] Center for Integrative Genomics, Faculty of Biology and Medicine, University of Lausanne, Génopode Building, Lausanne CH-1015, Switzerland. [2] Department of Neurobiology and Behavior, Cornell University, W153 Mudd Hall, Ithaca, NY 14853, USA. [3] Champalimaud Centre for the Unknown, Lisbon 1400-038, Portugal. [4] Department of Biology, Institute of Zoology, University of Fribourg, Chemin du Musée 10, Fribourg CH-1700, Switzerland. [5] Present address: Centre for Neural Circuits and Behaviour, University of Oxford, Tinsley Building, Mansfield Road, Oxford OX1 3SR, United Kingdom. These authors contributed equally: Ana Florencia Silbering, Vincent Croset, Giovanna Zappia. Correspondence and requests for materials should be addressed to R.B. (email: Richard.Benton@unil.ch)

Classic models of gustatory perception in mammals highlight the existence of a small number of taste classes signalling nutritive content (e.g. sugars and amino acids) or toxicity (e.g. bitter) that determine—through activation of hard-wired neural circuits—behavioural acceptance or rejection of food[1,2]. Different classes of tastants are recognised by discrete sensory channels that express distinct, and relatively small, receptor families. For example, detection of all sugars depends upon a single heterodimeric G protein-coupled receptor (GPCR) complex, T1R2/T1R3, while bitter cells—which detect an enormous diversity of noxious compounds—co-express a few dozen GPCRs of the T2R family[1,2].

Such models have been pervasive in interpreting how gustatory perception occurs in other animals, including insects, where analogous segregated sensory pathways for sweet and bitter compounds have been defined[3–6]. However, in contrast to mammals, where taste—mediated by lingual taste buds—informs only feeding decisions, insect gustation occurs in multiple sensory appendages, including the proboscis, legs, wings and sexual organs, and controls diverse behaviours, such as foraging, feeding, sexual/social recognition and oviposition[3–6]. In addition to stereotyped appetitive and aversive feeding responses to sweet and bitter compounds, respectively, insects display behavioural reactions to many other types of chemicals, including salt[7], water[8], carbonation (i.e. aqueous $CO_2$)[9], organic and inorganic acids[10,11], and pheromonal cuticular hydrocarbons[12].

The wide-ranging roles of the insect gustatory system are reflected in the molecular receptors that mediate peripheral sensory detection. The best-characterised taste receptor repertoire is the Gustatory Receptor (GR) family, which are a divergent set of presumed heptahelical ion channels that function in the detection of sugars, bitter compounds and certain sex pheromones[3,13]. A second large repertoire of receptors implicated in insect gustation is the Ionotropic Receptor (IR) family, which are ligand-gated ion channels that have derived from synaptic ionotropic glutamate receptors (iGluRs)[14–17]. Unlike iGluRs, IRs display enormous diversification both in the size of the repertoire across insects (ranging from tens to several hundreds[15,16,18]), and in their protein sequences (with as little as 10% amino acid identity between pairs of receptors). IRs are best-characterised in the vinegar fly, *Drosophila melanogaster*, which possesses 60 intact *Ir* genes. Of these, the most thoroughly understood are the 17 receptors expressed in the adult antenna. Thirteen of these are expressed in discrete populations of sensory neurons, and function as olfactory receptors for volatile acids, aldehydes and amines[16,19,20] or in humidity detection[21–24]. The remaining four (IR8a, IR25a, IR76b and IR93a) are expressed in multiple, distinct neuron populations and function, in various combinations, as co-receptors with the selectively-expressed tuning IRs[21,22,25].

By contrast, little is known about the sensory functions of the remaining, large majority of non-antennal IRs. Previous analyses described the expression of transgenic reporters for subsets of these receptors in small groups of gustatory sensory neurons (GSNs) in several different contact chemosensory structures[15,26–28]. While these observations strongly implicate these genes as having gustatory functions, the evidence linking specific taste ligands to particular receptors, neurons and behaviours remains sparse. For example, IR52c and IR52d are expressed in sexually-dimorphic populations of leg neurons and implicated in male courtship behaviours[26], although their ligands are unknown. Reporters for IR60b, IR94f and IR94h are co-expressed in pharyngeal GSNs that respond to sucrose, which may limit overfeeding[29] or monitor the state of externally digested food[30]. IR62a is essential for behavioural avoidance of high $Ca^{2+}$ concentrations, but the precise neuronal expression of this receptor is unclear[31]. As in the olfactory system, these selectively-expressed IRs are

likely to function with the IR25a and/or IR76b co-receptors, which are broadly-expressed in contact chemosensory organs, and required for detection of multiple types of tastants, including polyamines[32], inorganic, carboxylic and amino acids[28,33–35], and $Ca^{2+}$[31].

Here we describe a set of transgenic reporters for the entire *Ir* repertoire. We use these to survey the expression of this receptor family in both larval and adult stages. Using this molecular map, we identify IR56d as a selectively-expressed receptor that acts with IR25a and IR76b to mediate physiological and attractive behavioural responses to carbonation, a previously orphan taste class[9]. Furthermore, we extend recent studies[33,36,37] to show that IR56d is also required in sugar-sensing GR neurons to mediate distinct behavioural responses to fatty acids.

## Results

**A toolkit of transgenic reporters for IRs.** We generated transgenic reporters for all non-antennal IRs, comprising 5′ genomic regions of individual *Ir* loci placed upstream of Gal4 (Methods and Supplementary Table 1). Although the location of relevant gene regulatory sequences is unknown, this strategy has yielded faithful reporters of endogenous expression patterns for essentially all antennal *Irs*[14,20,21,38,39]. These constructs were integrated into identical sites in the genome to avoid positional effects on transgene expression. Such reporters of receptor expression provide greater sensitivity and spatial resolution than is permitted by RNA fluorescent in situ hybridization (FISH), which is inadequate to reliably detect *Ir* transcripts outside the antenna[14]. Moreover, when used to drive the membrane-targeted mCD8: GFP effector, they allow tracing of the innervation of labelled neurons in the brain and ventral nerve cord.

**IR neuronal expression and projections in larvae and adults.** We first analysed *Ir-Gal4* driven mCD8:GFP expression in third instar larvae (Figs. 1, 2 and Supplementary Fig. 1). In this analysis, we also included *Ir-Gal4* lines that are expressed in the adult antennae[20], and incorporated our previous observations on a small subset of the non-antennal IR reporters[15,28]. The larva contains a bilaterally-symmetric olfactory organ (dorsal organ) and several distinct gustatory organs located on the surface of the head and the internal lining of the pharynx (Figs. 1, 2)[40]. As described previously[27,28], the drivers for the co-receptors IR25a and IR76b (but not IR8a) are broadly expressed in all of these chemosensory organs (Figs. 1, 2). Expression of Gal4 drivers for only four other antennal IRs was detected in the dorsal organ: IR21a and IR93a, which act (with IR25a) in cool temperature-sensing[21,41], IR68a, which functions (with IR25a and IR93a) in moist air sensing[22,24] and IR92a, which mediates olfactory sensitivity to ammonia[14,42]. These observations suggest that the larval dorsal organ, like the adult antenna, has olfactory, thermosensory and hygrosensory roles.

Most reporters (27/44) of the remaining non-antennal IR repertoire are detected in bilaterally-symmetric populations of ~1–3 neurons in one or more larval gustatory organs, including head sensory neurons in the terminal and ventral organs, and internal neurons in the dorsal, ventral and posterior pharyngeal sense organs (Figs 1 and 2). Commensurate with these different peripheral expression patterns, the labelled neurons display diverse projection patterns in the primary gustatory centre, the subesophageal zone (SEZ) (Supplementary Fig. 1). Several reporters, for IR7d, IR7g, IR10a, IR68b and IR85a, are also detected in neurons in each segment of the abdomen, which project to the ventral nerve cord (VNC) (Fig. 1 and Supplementary Fig. 1).

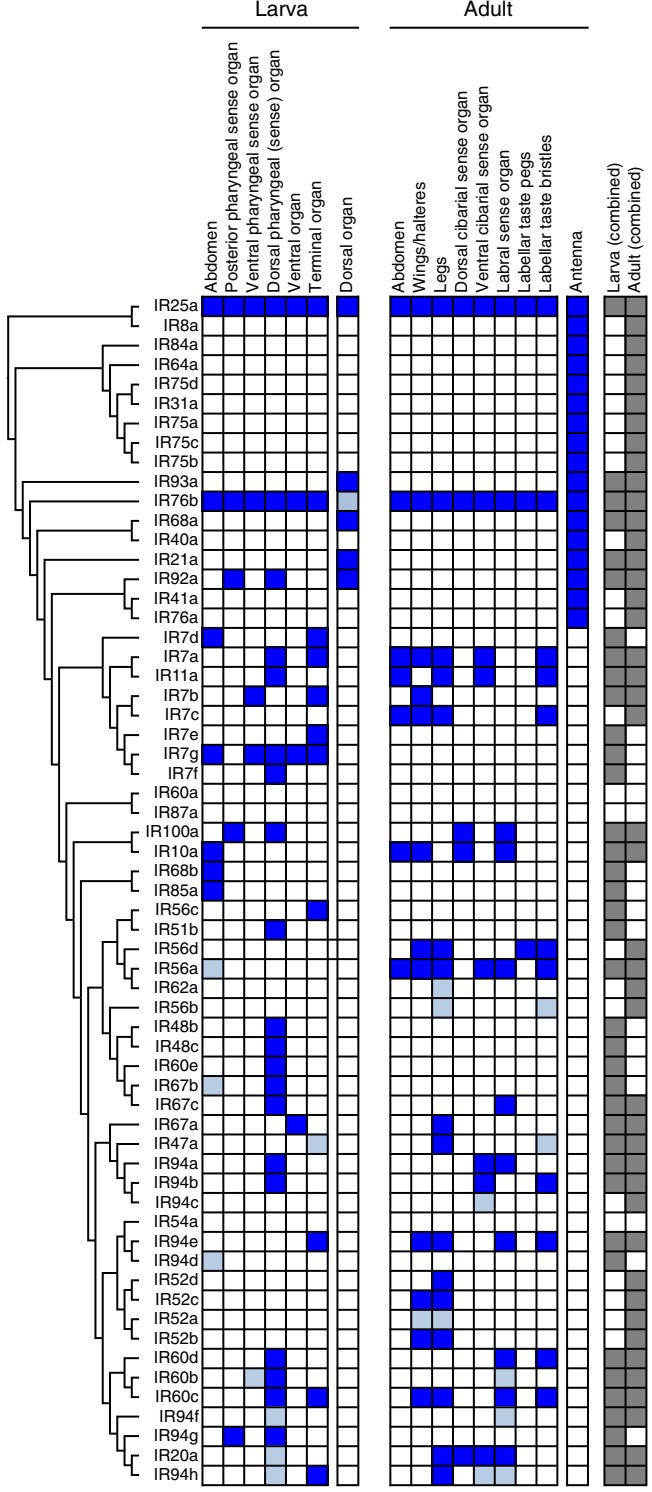

**Fig. 1** *Ir-Gal4* expression patterns and IR phylogeny. Summary of the expression (indicated by dark blue shading) of *Gal4* drivers for the entire *D. melanogaster* IR repertoire in peripheral chemosensory organs in third instar larvae and adult flies (see also Figs 2 and 3). Some lines, in particular antennal-expressed drivers, were previously described[15,20–22,25,28,39,41,74,75]. Light blue shading indicates additional expression reported for independently-generated *Ir-Gal4* drivers[26,27]. We did not distinguish expression in the dorsal pharyngeal organ and dorsal pharyngeal sense organ. The tree on the left shows a cladogram representing the phylogenetic relationships between *D. melanogaster* IRs. Protein sequences were aligned with MUSCLE, and the tree was made with RaxML under the WAG model of substitution, with 1000 bootstrap replicates. The columns on the right highlight drivers exhibiting common or stage-specific expression in larvae and adults

Supplementary Fig. 2). We noted sexually-dimorphic projection patterns in only two reporters: *Ir52c-Gal4* (similar to that previously described[26]) and *Ir94e-Gal4* (Supplementary Fig. 2); the latter driver also displays expression in a few soma within the SEZ (Supplementary Fig. 2).

**Relating receptor phylogeny, expression and life stage**. We combined these results with information on additional sites of expression revealed by a distinct set of reporters for a subset of IRs (the IR20a clade[26], which were built using 5' genomic regions of slightly different lengths as well as 3' sequences) to produce a global picture of *Ir* expression (Fig. 1). These data were organised by IR phylogeny, to examine the relationship between receptor protein sequences and spatiotemporal expression patterns. For the 44 non-antennal IRs, 32 reporters were expressed in larvae and 27 in adults, of which 17 were common to these life stages. Stage-specific receptors were found throughout the phylogeny (Fig. 1), rather than being confined to a single clade. Of the larval-specific IRs, nothing is currently known about their function; the adult-specific repertoire includes the *Ir52a-d* clade, some members of which control male mating behaviours[26].

In both life stages, drivers for some IRs that are closely-related in sequence (and often—but not always—encoded by tandemly-arrayed genes) are expressed in the same contact chemosensory organ (e.g. IR48b, IR48c, IR60e, IR67b and IR67c). This observation suggests that these more recently duplicated receptor genes retain similar *cis*-regulatory elements. However, this relationship is not strictly-held, as reporters for other, recently-diverged receptors can have quite different expression patterns (e.g. IR10a and IR100a).

**IR56d is expressed in labellar taste peg and bristle neurons**. To analyse the gustatory function of the non-antennal IRs, we focussed on IR56d, due to its unique expression: *Ir56d-Gal4* is the only reporter—apart from the broadly-expressed *Ir25a-Gal4* and *Ir76b-Gal4*—detected in neurons housed in the taste pegs, a class of short sensory hairs that lie between cuticular grooves (pseudotracheae) on the inner medial surface of the labellum (Fig. 4a, b). The driver is also expressed in neurons innervating taste bristles, which project from the external surface of the labellum (Fig. 4a, b). As we were unable to validate the expression of the *Ir56d-Gal4* transgene by RNA FISH, we used CRISPR/Cas9 genome editing to replace the endogenous *Ir56d* locus with *Gal4* to generate an independent driver line (*Ir56d^{Gal4}*) in which all relevant genomic regulatory regions should be present (Supplementary Fig. 3a). When combined with *UAS-mCD8:GFP*, *Ir56d^{Gal4}* displayed a highly similar expression pattern to the *Ir56d-Gal4* transgene

In adults, analysis of the new *Ir-Gal4* drivers did not identify any additional antennal-expressed IRs (Fig. 1). However, 21 reporters were detected within one or more populations of sensory neurons in external taste organs, including the taste bristles that project from the surface of the labellum, the labellar taste pegs, and the pharyngeal taste organs (Fig. 1 and Fig. 3). Furthermore, from examination of the central projections of these neurons to the SEZ and VNC, we surmised their expression in a variety of other taste organs, including the legs, wings, as well as neurons that may originate in the abdomen (Fig. 1 and

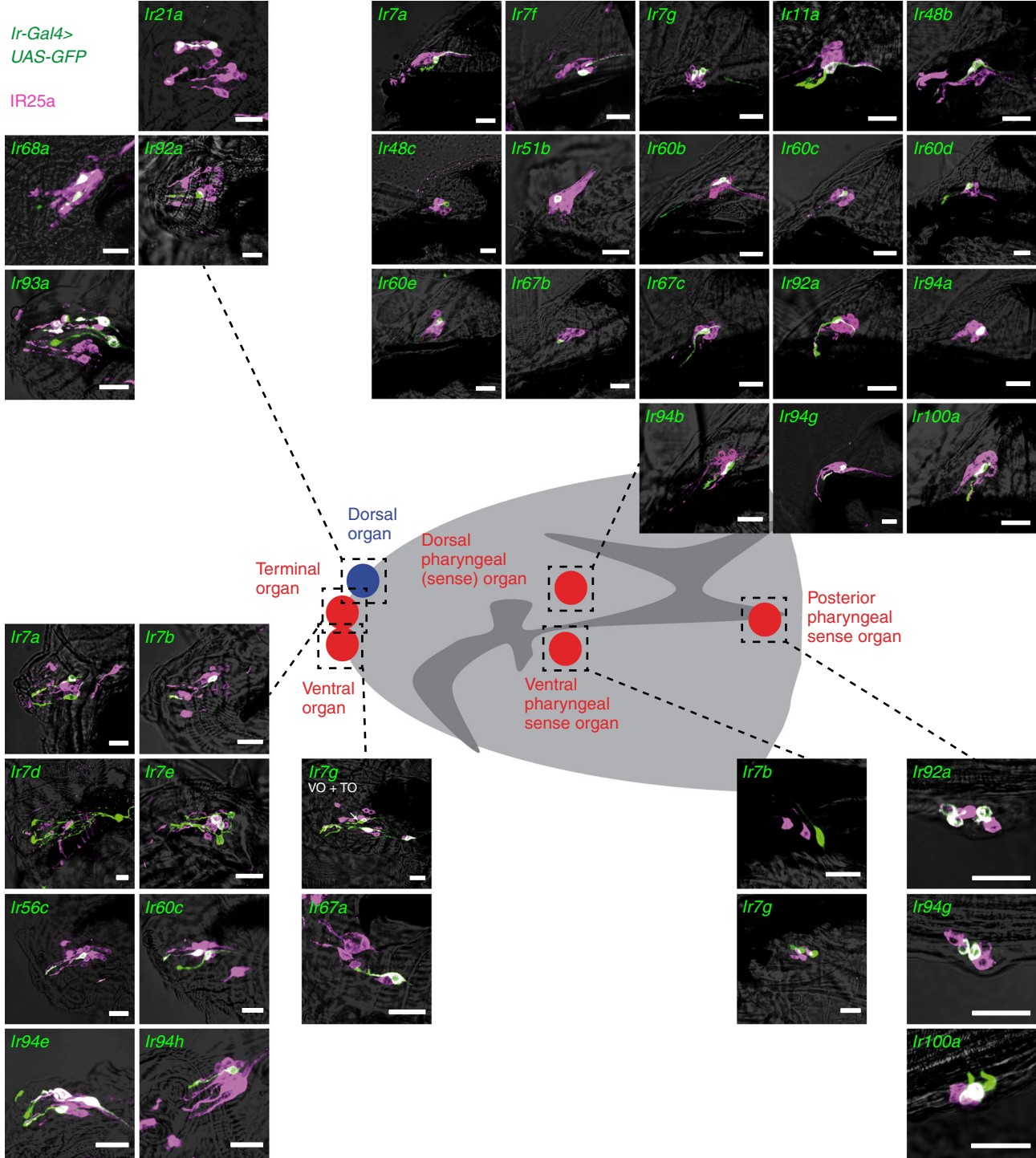

**Fig. 2** Expression of *Ir-Gal4* drivers in the peripheral nervous system of larval *Drosophila*. Immunofluorescence with anti-GFP (green) and anti-IR25a (magenta), overlaid on bright-field images, on whole-mount tissue of third instar larvae, revealing the expression of *Ir-Gal4* lines in different chemosensory organs (as schematised in the cartoon of the larval head in the centre). *Ir7g-Gal4* is expressed in both terminal organ (TO) and ventral organ (VO) neurons; the VO neuron is indicated with an arrow. Images for *Ir7b*, *Ir7e*, *Ir7g*, *Ir56c*, *Ir60c* and *Ir94e* drivers are adapted from[28]. Genotypes are of the form: *w;UAS-mCD8:GFP;IrX-Gal4*. Scale bars: 20 μm

(Supplementary Fig. 3b), indicating that the original promoter reporter faithfully recapitulates endogenous gene expression.

To characterise the identity of the IR56d neurons, we combined the *Ir56d-Gal4* driver (or an equivalent *Ir56d-LexA* transgene; see Methods) with reporters for other populations of labellar neurons. We first confirmed that IR56d neurons express IR25a and IR76b (Fig. 4c), suggesting that IR56d may function

with one or both of these co-receptors. Morphological studies have shown that taste pegs contain one presumed mechanosensory and one chemosensory neuron[43]. The mechanosensory neuron can be visualised with a promoter reporter for the NOMPC mechanoreceptor (*NompC-LexA*)[44,45]. We observed that *NompC-LexA*-labelled neurons paired, but did not overlap, with *Ir56d-Gal4*-positive taste peg neurons (Fig. 4d). By contrast,

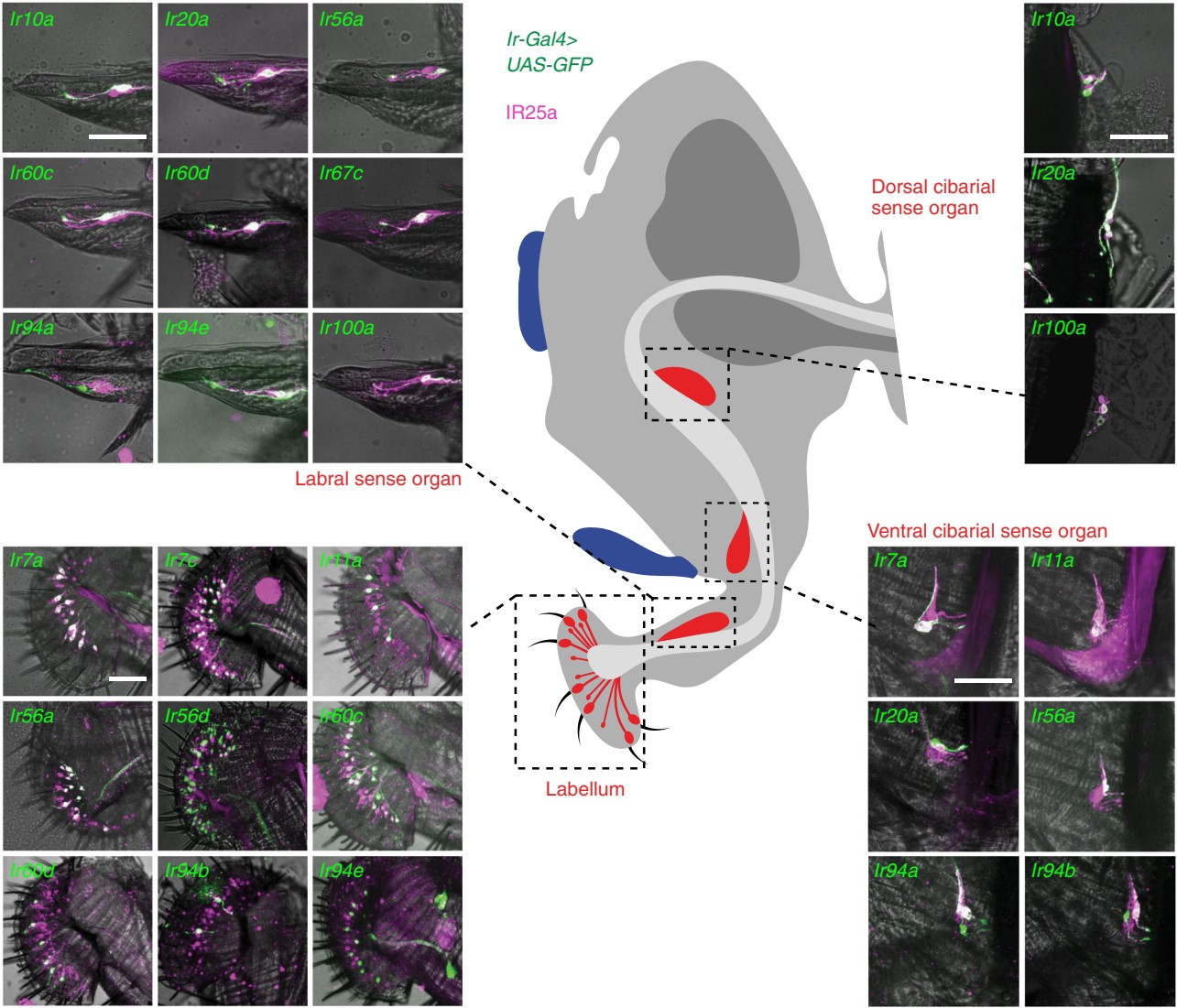

**Fig. 3** Expression of *Ir-Gal4* drivers in the proboscis of adult *Drosophila*. Immunofluorescence with anti-GFP (green) and anti-IR25a (magenta), overlaid on bright-field images, on whole-mount proboscides revealing the expression of *Ir-Gal4* lines in different adult gustatory organs (as schematised in the cartoon in the centre). Genotypes are of the form: *w;UAS-mCD8:GFP;IrX-Gal4*. Scale bars: 50 μm

*Ir56d-Gal4*-expressing cells in the taste pegs co-localised with those labelled by the *E409-Gal4* enhancer trap, which labels at least a subset of the peg chemosensory neurons[9] (Fig. 4e). Taste bristles house two to four gustatory neurons, including those tuned to sweet and bitter stimuli, labelled by reporters for *Gr64f* and *Gr66a*, respectively[3,6]. *Ir56d-Gal4* taste bristle neurons were completely distinct from *Gr66a*-positive cells, but overlapped with a subset of the *Gr64f*-expressing neurons (Fig. 4f, g).

Consistent with the expression in pegs and bristles, *Ir56d-Gal4* neurons project to two distinct regions of the SEZ: the anterior maxillary sensory zone 1 (AMS1), and the posterior maxillary sensory zone 4 (PMS4) (Fig. 4h)[46]. Co-labelling of these neurons with the *Gr64f* reporter demonstrated that the taste bristle subpopulation innervates PMS4, indicating that the taste peg neurons project to AMS1 (Fig. 4h), consistent with previous observations[9,46].

**IR56d taste peg neurons are gustatory carbonation sensors**. To determine the physiological specificity of IR56d neurons, we expressed the fluorescent calcium indicator GCaMP3 under the control of *Ir56d-Gal4* (Fig. 5a), and measured changes in fluorescence in their axon termini in the SEZ upon presentation to the proboscis of a panel of diverse taste stimuli, including sugars, bitter compounds, amino and organic acids, high and low NaCl concentrations, carbonated solutions and buffers of different pH (Fig. 5b, c). We separately quantified GCaMP3 fluorescence changes in the AMS1 and PMS4 projections, reflecting activity of taste peg and taste bristle subpopulations, respectively. AMS1-innervating neurons responded strongly to carbonated solutions (Fig. 5c), but not to other tastants in this panel. These data—together with our co-expression analysis (Fig. 4e)—identify the *Ir56d* taste peg neurons as the carbonation-sensing cells that were previously recognised by their expression of the *E409-Gal4* enhancer trap[9].

PMS4-innervating neurons displayed a broader response profile, showing the largest GCaMP3 fluorescence changes upon stimulation with sucrose and other sugars, consistent with these neurons representing a subset of the *Gr64f*-expressing sweet-sensing neurons housed in taste bristles (Fig. 4g). We also detected weaker responses to glycerol, acetic acid, and, somewhat variably, to carbonated solutions (Fig. 5c). These observations

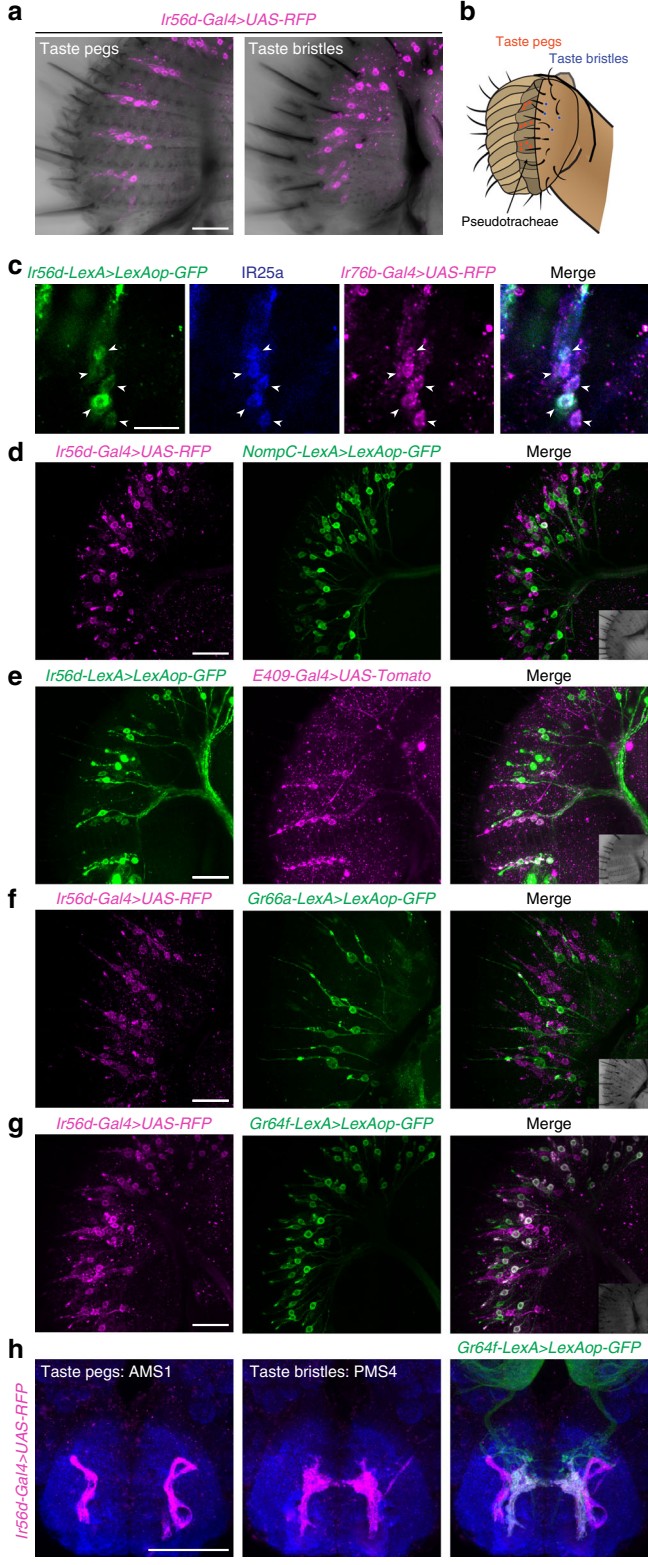

**Fig. 4** IR56d is expressed in two populations of neurons in the labellum. **a** Immunofluorescence with anti-RFP (magenta), overlaid on bright-field images, on a whole-mount proboscis of a *w;UAS-mCD8:RFP;Ir56d-Gal4* animal. The left image corresponds to the maximal projection of the inner face of one labellar palp, and the right image corresponds to the surface of one labellar palp. Scale bar: 25 μm. **b** Schematic representing the anatomical location of the taste peg neurons (orange) and taste bristle neurons (blue) in the labellum. **c** Immunofluorescence with anti-GFP (green), anti-IR25a (blue) and anti-RFP (magenta) on a whole-mount proboscis of a *w;LexAop-mCD8:GFP-2A-mCD8:GFP/UAS-mCD8:RFP;Ir56d-LexA/Ir76b-Gal4* animal. The images show a close-up of taste peg neurons (arrowheads) to visualise the co-localisation of the three markers. Scale bar: 25 μm. **d** Immunofluorescence with anti-RFP (magenta) and anti-GFP (green) on a whole-mount proboscis of a *w;LexAop-mCD8:GFP-2A-mCD8:GFP/UAS-mCD8:RFP;NompC-LexA/Ir56d-Gal4* animal. The inset in the merged image shows a bright-field view of the imaged tissue (here and in the following panels). Scale bar: 25 μm. **e** Immunofluorescence with anti-RFP (magenta) and anti-GFP (green) on a whole-mount proboscis of a *w;LexAop-mCD8:GFP-2A-mCD8:GFP/E409-Gal4;Ir56d-LexA/UASCD4:tdTomato* animal. Scale bar: 25 μm. **f** Immunofluorescence with anti-RFP (magenta) and anti-GFP (green) on a whole-mount proboscis of a *Gr66a-LexA/+; LexAop-rCD2:GFP/UAS-mCD8:RFP;Ir56d-Gal4/(TM6B or TM2)* animal. Scale bar: 25 μm. **g** Immunofluorescence with anti-RFP (magenta) and anti-GFP (green) on a whole-mount proboscis of a *w;LexAop-mCD8:GFP-2A-mCD8:GFP/UAS-mCD8:RFP;Gr64f-LexA/Ir56d-Gal4* animal. Scale bar: 25 μm. **h** Immunofluorescence with anti-RFP (magenta), anti-GFP (green) and nc82 (blue) on a whole-mount brain of a *w;LexAop-mCD8:GFP-2A-mCD8:GFP/UAS-mCD8:RFP;Gr64f-LexA/Ir56d-Gal4* animal. Both left and middle panels show the expression of only the *Ir56d-Gal4* driver. The left panel shows the maximal projection of the anterior SEZ; the middle panel shows the maximal projection of the most posterior optical slices of the SEZ. The right panel shows the overlay of the *Ir56d-Gal4*- and *Gr64f-LexA*-expressing populations. AMS1 anterior maxillary sensory zone 1, PMS4 posterior maxillary sensory zone 4. Scale bar: 50 μm

type of IR56d neuron, which is equivalent to that housed in labellar taste bristles.

**IR56d, IR25a and IR76b are required for carbonation sensing.** To address the contribution of IR56d to the sensory responses of the neurons in which it is expressed, we used CRISPR/Cas9 genome editing to generate two *Ir56d* mutant alleles; these contain frame-shift generating deletions predicted to truncate the protein within the presumed ligand-binding domain (*Ir56d[1]*) or before the ion channel domain (*Ir56d[2]*) (Fig. 6a). We performed calcium imaging in IR56d neurons in *Ir56d* mutant flies using sucrose and carbonation stimuli, which were the strongest agonists for the taste bristle (PMS4) and taste peg (AMS1) subpopulations, respectively (Fig. 5c). While responses of the mutants to sucrose were unaffected compared to control animals, responses to carbonation were abolished in *Ir56d* mutants (Fig. 6b, c). The defect in sensitivity to carbonation was restored upon selective expression of a wild-type *Ir56d* cDNA in these neurons (Fig. 6c).

We next tested the contribution of the two other IRs expressed in IR56d neurons, IR25a and IR76b. Mutations in each of these genes produced phenotypes that were very similar to those of *Ir56d* mutants: sucrose responses in the PMS4 were unaffected, while responses to carbonation were completely lost (Fig. 6b, c). Sensitivity to carbonation could be rescued by expression of wild-type cDNA transgenes in the corresponding mutant background (Fig. 6b, c). Together these data suggest that the carbonation sensor comprises, at least in part, a complex of IR56d with the co-receptors IR25a and IR76b. The persistent sucrose responses in

indicate that *Ir56d* subpopulations are both anatomically and physiologically distinct.

*Ir56d-Gal4* is additionally expressed in leg neurons (Fig. 1 and Supplementary Fig. 2); consistent with previous observations[33], these cells are also labelled with a *Gr64f* reporter (Supplementary Fig. 4a). By calcium imaging in leg tarsi, we found these cells respond to sucrose but not carbonated solutions (Supplementary Fig. 4b-c). These results suggest that the legs contain only one

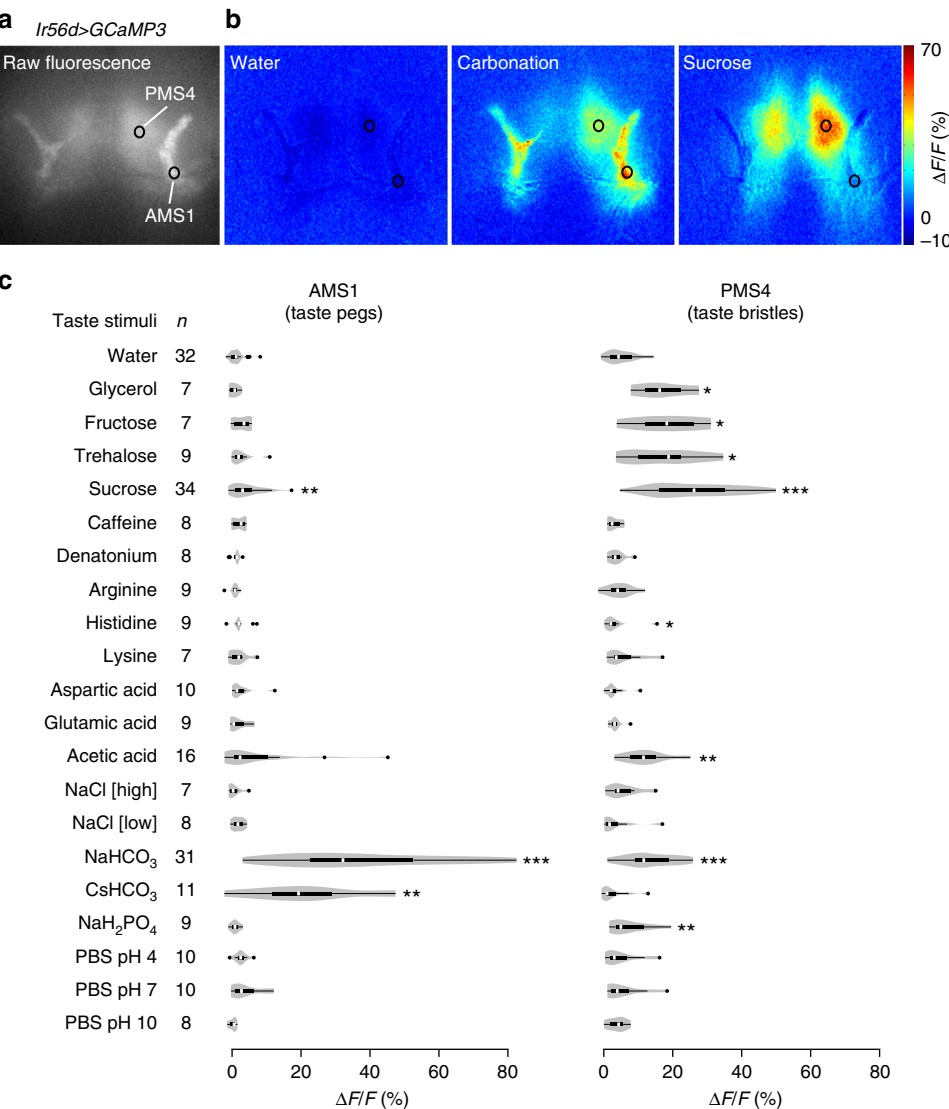

**Fig. 5** IR56d neurons respond to carbonation and sugars. **a** Raw fluorescence image of IR56d neuron axons innervating the SEZ in a *w;UAS-GCaMP3;Ir56d-Gal4* animal. The black circles indicate the approximate position of the regions-of-interest (ROIs) in the AMS1 and PMS4 used for the quantifications in **c**. **b** Colour-coded images of $\Delta F/F$ (reflecting the maximal GCaMP3 fluorescence intensity changes; scale bar on the far-right) of IR56d neuron responses in the same animal as in **a** to the application of distilled water, a carbonated solution or sucrose to the labellum. **c** Quantification of changes in $\Delta F/F$ in the ROIs in AMS1 and PMS4 upon application of the indicated taste stimuli to the labellum of *w;UAS-GCaMP3;Ir56d-Gal4* animals. Violin plots (in this and all following figures) show the density of the data points extending from the minimum to the maximum value. Internal boxplots show the median and the interquartile range of the distribution of responses (IQR; first and third quartile). Whiskers represent 1.5x IQR. Black dots represent outliers. *n* denotes the number of replicates for each stimulus. Concentrations of each of the taste stimuli are listed in Supplementary Table 2. For the statistical analysis the response data for each stimulus were compared with water; only significant differences are shown: *$P < 0.05$, **$P < 0.01$, ***$P < 0.001$ (Wilcoxon rank sum test)

*Ir56d* taste bristle neurons lacking these IRs are consistent with the well-established role of GRs in sugar sensing in these cells[3,6].

**Carbonation induces *Ir56d*-dependent attraction.** Previous analysis of flies' behavioural responses to carbonation indicated that this stimulus mediates taste-acceptance behaviour[9]. However, the requirement of *E409-Gal4*-positive labellar neurons was difficult to determine because the *E409-Gal4* enhancer trap is expressed in many central neurons in addition to the taste pegs[9], limiting its usefulness for neuronal manipulation experiments. With our characterisation of IR56d, we were better positioned to examine more precisely the sensory basis of carbonation-evoked behaviours.

We first established a two-choice assay in which flies could freely explore a circular arena containing separate semicircles of filter paper soaked in carbonated or non-carbonated solutions (100 mM NaHCO₃ pH 6.5 and 100 mM NaHCO₃ pH 8.5, respectively; these ensure a long-lasting source of carbonation[9]). After 90 min, we observed the position of flies in the arena and calculated a preference index (Fig. 7a). Wild-type flies showed a clear preference for the carbonated solution (Fig. 7b). This preference was not due to the pH difference of the solutions as flies did not show preference for phosphate buffered saline (PBS) pH 6.5 over PBS pH 8.5 (Fig. 7c); similarly, the slightly different salt concentrations in the carbonated and non-carbonated solutions (see Methods) could not account for the preference

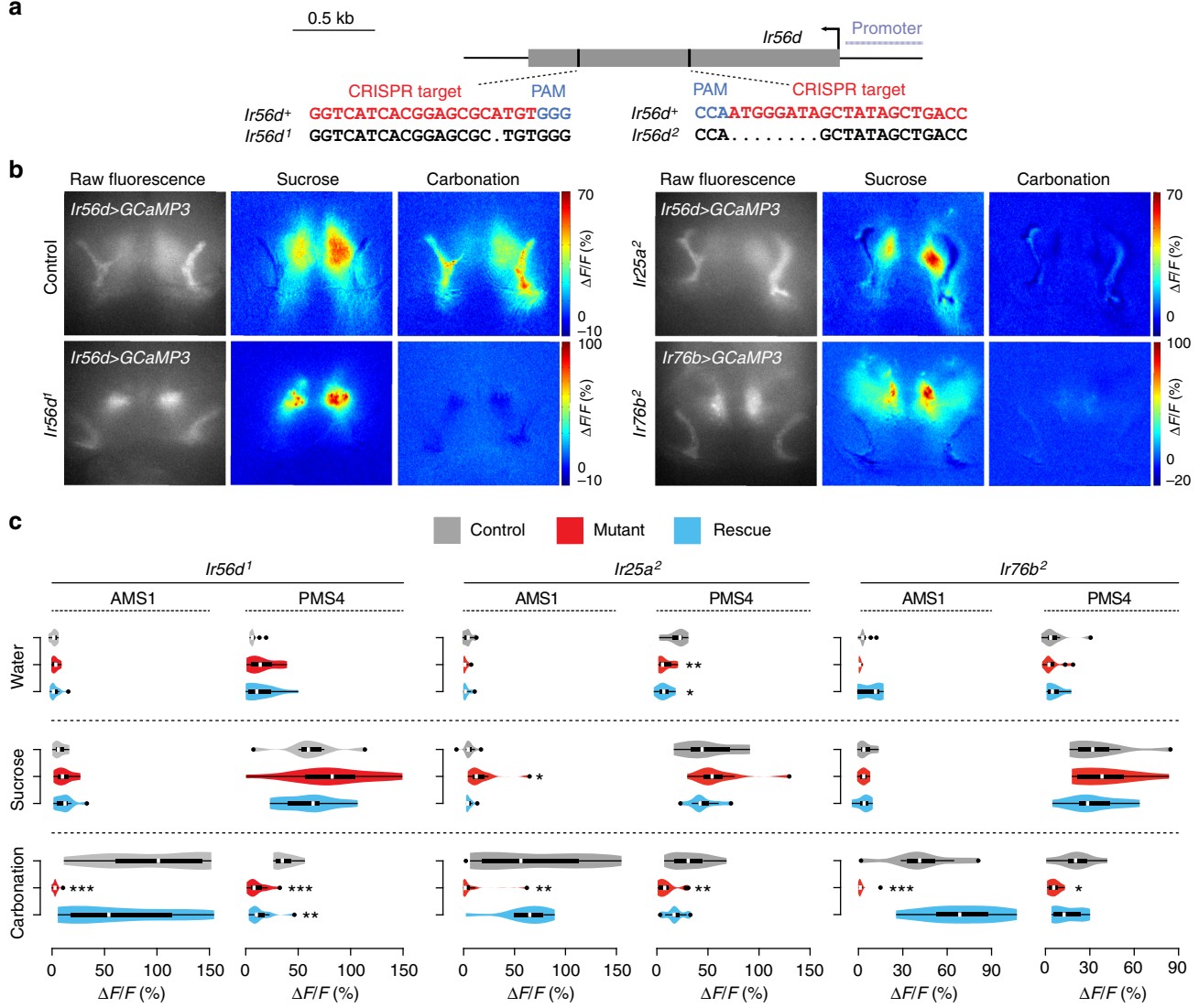

**Fig. 6** IR56d, IR25a and IR76b are required for sensory responses to carbonation but not sucrose. **a** Schematic of the *Ir56d* locus (single exon in grey), showing the position of the CRISPR target and the sequence of the *Ir56d* mutant alleles in these regions. PAM protospacer adjacent motif. **b** Raw fluorescence of GCaMP3 expressed in IR56d neurons, and colour-coded images (reflecting the maximal GCaMP3 fluorescence intensity changes; scale bars for each genotype on the right) in control, *Ir56d¹*, *Ir25a²* and *Ir76b²* mutant flies stimulated with 1 M sucrose and a carbonated solution (as in Fig. 5c). For genotypes, see **c**. **c** Quantification of changes in ΔF/F in the ROIs as shown in Fig. 5a, b upon application of the indicated chemicals to the proboscis of the indicated genotypes: IR56d: Control: *w;Bl/+; UAS-GCaMP3,Ir56d-Gal4/+* (n = 8); Mutant: *w;Ir56d¹/Ir56d¹; UAS-GCaMP3,Ir56d-Gal4/+* (n = 11); Rescue: *w;Ir56d¹,UAS-Ir56d/Ir56d¹;UAS-GCaMP3,Ir56d-Gal4/+* (n = 10). IR25a: Control: *w;Bl/+; UAS-GCaMP3,Ir56d-Gal4/+* (n = 11); Mutant: *w;Ir25a²/Ir25a²;UAS-GCaMP3,Ir56d-Gal4/+* (n = 11); Rescue: *w;Ir25a²,UAS-Ir25a/Ir25a²; UAS-GCaMP3,Ir56d-Gal4/+* (n = 9). IR76b: Control: *w;Ir76b-Gal4/CyO;UAS-GCaMP3/TM6B* (n = 8); Mutant: *w;Ir76b-Gal4/+; UAS-GCaMP3,Ir76b²/Ir76b²* (n = 10); Rescue: *w;Ir76b-Gal4,UAS-Ir76b/+;UAS-GCaMP3,Ir76b²/Ir76b²* (n = 7). We used *Ir76b-Gal4* in the rescue experiments of the *Ir76b* mutant due to constraints of the chromosomal location of the relevant transgenes; although *Ir76b-Gal4* is more broadly-expressed than *Ir56d-Gal4*, the AMS1 and PMS4 projections are still easily recognisable. For the statistical analysis the response data for each stimulus were compared with water; only significant differences are shown: \*P < 0.05, \*\*P < 0.01, \*\*\*P < 0.001 (Wilcoxon rank sum test with Bonferroni correction for multiple comparisons)

observed (Fig. 7d). These observations are consistent with those made using a different positional-preference assay[9], confirming that carbonation (a product of microbial fermentation) is a modestly attractive stimulus for *Drosophila*. Importantly, this preference was completely abolished in *Ir56d* mutant flies (Fig. 7b) and restored, albeit not to wild-type levels, by expression of *Ir56d* cDNA in IR56d neurons (Fig. 7b).

To investigate why flies display positional preference for carbonation, we performed additional behavioural assays. The best-established response of insects to attractive gustatory stimuli

is the proboscis extension reflex (PER), which promotes contact of the feeding organ with the substrate. While PER is robustly triggered by sucrose (Fig. 7e), the carbonated solution used in the two-choice assay (100 mM NaHCO₃ pH 6.5) triggered a small PER response that was only slightly higher than the control non-carbonated solution (100 mM NaHCO₃ pH 8.5) (Fig. 7e). To eliminate any contribution of salt-evoked PER, we also performed PER assays with fresh commercial carbonated and non-carbonated water, which have only trace levels of minerals (Supplementary Table 2). Here, both stimuli induced similarly

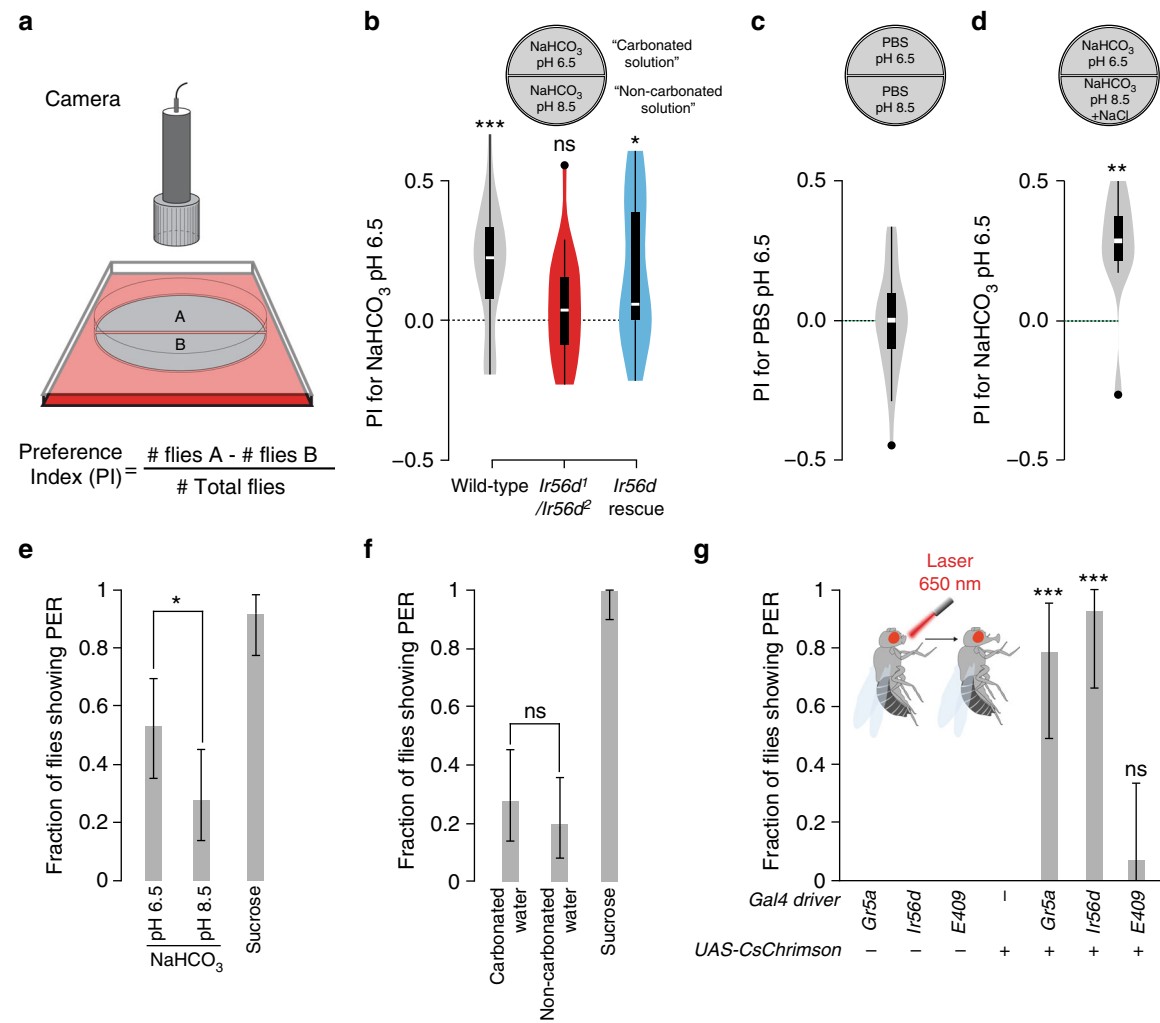

**Fig. 7** Carbonation promotes IR56d-dependent feeding behaviours: **a** Schematic of the two-choice positional preference arena assay. Flies can choose to feed from two substrates comprising stacks of blotting paper soaked in different tastant solutions on opposite sides of a 94 mm diameter Petri dish. Fly position was quantified automatically and used to calculate a Preference Index (PI) as indicated below the scheme. **b** Two-choice assay showing the preference of the indicated genotypes for a carbonated solution (100 mM NaHCO$_3$ pH 6.5) over a control non-carbonated solution (100 mM NaHCO$_3$ pH 8.5). Genotypes: $w^{1118}$ ($n = 21$ arenas; 70–80 flies per arena); $Ir56d$ mutant: $w;Ir56d^1/Ir56d^2$ ($n = 21$ arenas); Rescue: $w;Ir56d^1,UAS-Ir56d/Ir56d^2;Ir56d-Gal4/$ $+$ ($n = 14$ arenas). ns non-significant, *$P < 0.05$, **$P < 0.01$, ***$P < 0.001$ (Wilcoxon signed rank test ($H_0 = 0$)). **c** Two-choice assay showing the preference of $w^{1118}$ flies ($n = 24$ arenas) for a PBS pH 6.5 solution vs. a PBS pH 8.5 solution. **d** Two-choice assay showing the preference of $w^{1118}$ flies ($n = 10$ arenas) for a carbonated solution (100 mM NaHCO$_3$ pH 6.5) over a non-carbonated solution (100 mM NaHCO$_3$ pH 8.5) that was supplemented with NaCl to achieve a final [Na$^+$] of 150 mM. **$P < 0.01$ (Wilcoxon signed rank test ($H_0 = 0$)). **e** Fraction of $w^{1118}$ flies ($n = 36$) showing the proboscis extension reflex (PER) to the tastants indicated (100 mM NaHCO$_3$ at pH 6.5 or pH 8.5, 100 mM sucrose). Error bars represent the ±95% binomial confidence intervals; *$P < 0.05$ (Fisher exact test). **f** Fraction of $w^{1118}$ flies ($n = 36$) showing PER to commercial carbonated water, non-carbonated water and 100 mM sucrose. Error bars represent the ±95% binomial confidence intervals; ns non-significant (Fisher exact test). **g** Fraction of flies of the indicated genotypes ($n = 14$ for all) showing PER upon optogenetic stimulation using red light. Genotypes (left-to-right along the x-axis): (i) $w;Bl/CyO;Gr5a-Gal4/TM6B$ (ii) $w;Bl/CyO;Ir56d-Gal4/TM6B$ (iii) $w;Bl/CyO;E409-Gal4/TM6B$ (iv) $w;UAS-CsChrimson/CyO;TM2/TM6B$ (v) $w;UAS-CsChrimson/CyO;Gr5a-Gal4/TM6B$ (vi) $w;UAS-CsChrimson/CyO;Ir56d-Gal4/TM6B$ (vii) $w;UAS-CsChrimson/CyO;E409-Gal4/TM6B$. Error bars represent the ±95% binomial confidence intervals; ns non-significant, ***$P < 0.001$ (Fisher exact test)

low levels of PER (Fig. 7f). Finally, we examined whether PER can be triggered by optogenetic activation of taste peg neurons using the red-light sensitive channelrhodopsin CsChrimson (Fig. 7g). In positive control animals, in which CsChrimson was expressed under the control of a *Gr5a* driver or our *Ir56d* driver (which is expressed in both taste pegs and sugar-sensing neurons in taste bristles), exposure of the labellum to red light induced, as expected, robust PER (Fig. 7g). By contrast, selective activation of the taste peg neurons (using the *E409-Gal4* driver) did not (Fig. 7g). Together these results suggest that the carbonation-sensing taste peg neurons do not activate the PER motor circuit.

During the positional preference assay, flies might also taste the substrate with their legs. However, direct carbonation stimulation of legs does not evoke PER (Supplementary Fig. 4d), which is consistent with the lack of physiological sensitivity of tarsal neurons to this stimulus (Supplementary Fig. 4b-c). These observations argue that the IR56d leg neurons are unlikely to contribute to the behavioural responses to carbonation, and that labellar taste peg neurons are the principal (and potentially only) carbonation sensors in the animal.

Finally, we asked whether carbonation influences food ingestion using Expresso, an automated feeding assay that can

measure the number and volume of individual meal-bouts[47]. When comparing feeding of wild-type flies on carbonated and control solutions, we found no difference in any of the parameters measured (Supplementary Fig. 5a). However, we noted that these stimuli were very poor inducers of feeding, with fewer than half the flies consuming very low volumes of solutions. We reasoned this was due to the lack of a nutritious substance, and repeated the assays in the presence of a low concentration of sucrose (5 mM), which is moderately attractive to Drosophila[48]. This sugar supplement greatly increased consumption by the flies, but we again did not observe any enhancement of feeding by carbonation (Supplementary Fig. 5b-c). Thus, the attractiveness of carbonation to flies does not appear to be related to an ability to induce increased consumption, at least in this assay and with the tested conditions.

**Ir56d-dependent hexanoic acid sensing in taste bristles**. In the course of completion of our study, the taste bristle neurons that co-express Ir56d and sweet-sensing Grs in the labellum and legs were found to mediate physiological and behavioural responses to medium chain fatty acids[33,36]. We confirmed these observations by showing that hexanoic acid activates IR56d neurons, noting that the strongest responses occur in taste peg neurons (Supplementary Fig. 6a-b). Hexanoic acid responses were abolished in Ir56d mutants (Supplementary Fig. 6a-b), suggesting that IR56d functions both in carbonation and fatty acid detection. In contrast to carbonation, however, fatty acids evoke PER, and this behaviour is abolished in Ir56d mutants (Supplementary Fig. 6c). As taste peg neuron activation does not trigger PER (Fig. 7e–g), these observations suggest that hexanoic acid-evoked activity in taste bristles is responsible for this behaviour, as proposed previously[36]. Consistent with this hypothesis, RNAi of Ir56d specifically in the sweet-sensing Gr neuron subpopulation eliminates fatty acid-induced PER[33].

## Discussion

This work describes the first family-wide survey of the expression of IRs in Drosophila, revealing remarkable diversity in the neuronal expression patterns of members of this repertoire across all known chemosensory organs in both larvae and adults. These observations reinforce previous conclusions from analyses of subsets of these genes[15,26–28] that the non-antennal IRs function to detect a myriad of chemical stimuli to evoke a variety of behavioural responses. Such properties presumably apply to the vast, divergent IR repertoires of other insect species[15], for example, the 455 family members in the German cockroach Blatella germanica[18], or the 135 IRs in the mosquito Aedes aegypti[49]. Within Drosophila we did not detect obvious relationships between IR phylogeny and stage- or organ-specific expression patterns. Phylogenetic proximity may therefore be the most indicative of functional relationships between IRs, as is the case for those expressed in the antenna[20]. If this hypothesis is correct, the expression data presented here suggest that functionally-related clades of receptors act in several types of chemosensory organ.

An important caveat to the transgenic approach we used to reveal expression is the faithfulness of these reporters to the endogenous expression pattern of Ir genes. Although this strategy has been widely (and successfully) used for antennal IRs and other chemosensory receptor families, it is impossible to determine reporter fidelity without a complementary tool (e.g. receptor-specific antibodies or tagging of the endogenous genomic locus). We note discrepancies between the expression of some of our Ir-Gal4 lines and those described previously[26]; many of these probably reflect differences in the length of regulatory

regions used to create these distinct transgenes. Precise comparison of independently-constructed transgenic constructs may in fact be useful in informing the location of enhancer elements directing particular temporal or spatial expression patterns. Moreover, transgenic reporters provide powerful genetic tools for visualisation and manipulation of specific neuronal populations. The reagents generated here should therefore provide a valuable resource for further exploration of the IRs in insect gustation.

Using our atlas, we identified IR56d—together with the broadly-expressed co-receptors IR25a and IR76b—as essential for responses of labellar taste peg neurons to carbonation. Such observations implicate IR56d as the previously unknown tuning receptor for this stimulus[9]. However, these IRs do not appear to be sufficient for carbonation detection, as their misexpression in other neurons failed to confer sensitivity to carbonated stimuli (Supplementary Fig. 7). This observation suggests that additional molecules or cellular specialisations are required. Such a factor may be rather specific to taste pegs, given the minimal/absent responses of Ir56d-expressing taste bristle/leg neurons to carbonation, but does not appear to be another IR, as we have not identified other IR reporters expressed in this population of cells.

While precise mechanistic insights into carbonation sensing will require the ability to reconstitute IR56d-dependent carbonation responses in heterologous systems, it is interesting to compare how insects and mammals detect this stimulus. The main mammalian gustatory carbonation sensor, the carbonic anhydrase Car4[50] is an enzyme tethered to the extracellular surface of sour (acid) taste receptor cells in lingual taste buds, where it is thought to catalyse the conversion of aqueous $CO_2$ into hydrogencarbonate (bicarbonate) ions ($HCO_3^-$) and protons ($H^+$). The resulting free protons, but not hydrogencarbonate ions, provide a relevant signal for the sour-sensing cells[50]. By contrast, IR56d neurons are not responsive to low pH, suggesting a different chemical mechanism of carbonation detection. Our observation that IR56d is also essential for sensitivity to hexanoic acid suggests that IR56d could recognise the common carboxyl group of hydrogencarbonate and fatty acid ligands. However, IR56d neurons are not responsive to all organic acids, indicating that this cannot be the only determinant of ligand recognition.

Our characterisation of IR56d neurons extends previous reports[33,36,37] to reveal an unexpected complexity in the molecular and neuronal basis by which attractive taste stimuli are encoded. The taste bristle population of IR56d neurons represents a subset of sugar-sensing cells that are also responsive to fatty acids, glycerol and, minimally, to carbonation. Although activation of these neurons promotes PER, we find that carbonation-evoked stimulation is insufficient to trigger this behaviour, which suggests that taste bristles are not a relevant sensory channel for this stimulus. While members of a specific clade of GRs are well-established to mediate responses to sugars and glycerol[3,6,37,51], the detection mechanisms of fatty acids appear to be more complex. Earlier work demonstrated an important role of a phospholipase C homologue (encoded by norpA) in labellar fatty acid responses[10]. More recently, GR64e was implicated as a key transducer of fatty acid-dependent signals, but suggested to act downstream of NorpA, rather than as a direct fatty acid receptor[37]. By contrast, an independent study of the legs showed that all sugar-sensing Gr genes (including Gr64e) were dispensable for fatty acid detection, and provided evidence instead for an important role of IR25a and IR76b in these responses[33]. Analysis of our Ir56d mutants indicates an IR-dependent fatty acid-detection mechanism also exists in the labellum; future work will be needed to relate this to the roles of GR64e and NorpA.

The IR56d taste peg population is, by contrast, sensitive to carbonation and fatty acids (but not sugars or glycerol), and these responses can be ascribed to IR56d (a Gr64e[LexA] reporter is not

expressed in taste peg neurons[52]). Although these neurons mediate taste-acceptance behaviour, they do not appear to promote proboscis extension or food ingestion. Recent work using optogenetic neuronal silencing experiments provided evidence that taste peg neuron activity is important for sustaining, rather than initiating, feeding on yeast, by controlling the number of sips an animal makes after proboscis extension[53]. These observations are concordant with the internal location of taste pegs on the labellum, as they will not come into contact with food until the proboscis has been extended, and could explain the positional preference for carbonated substrates that we observed. We have attempted to determine whether carbonation can influence sipping behaviour using flyPAD[54]. Although these experiments did not reveal a statistically-significant effect (Supplementary Fig. 8), interpretation is complicated by the difficulty of providing and maintaining carbonation stimuli in the solid medium used in flyPAD assays. Future development of other approaches to provide this stimulus in feeding assays will be necessary. Nevertheless, our data strengthen the view that carbonation, a non-nutritious microbial fermentation product, regulates—via activation of IR56d taste peg neurons—a distinct motor programme to PER as part of a multicomponent behavioural response.

## Methods

**Transgene generation**. *Ir-Gal4* lines were constructed with sequences from the Oregon R strain (OR) using standard methods[15,28] (Supplementary Table 1) and inserted into the *attP2* landing site[55], by normal transformation procedures (Genetic Services, Inc.). *Ir56d-LexA* was made by subcloning the same genomic sequence as in *Ir56d-Gal4* upstream of *LexA:VP16-SV40*[56] in *pattB*[57] and transformation into *attP2*. *UAS-Ir56d* was made by PCR amplification of the *Ir56d* (single-exon) ORF from *w1118* genomic DNA, which was T:A cloned into pGEM-T Easy and sequenced, before subcloning into *pUAStattB*[57], and transformation into *attP40*[55]. *UAS-Ir56dmut* contains a deletion of a single nucleotide at position 1010 of the *Ir56d* ORF; this frameshift mutation is predicted to cause premature translation termination and a non-functional receptor fragment.

**Drosophila strains**. *Drosophila* stocks were maintained on a standard corn flour, yeast and agar medium under a 12 h light:12 h dark cycle at 25 °C; different culture conditions for certain behavioural assays are described below. The wild-type strain was *w1118*. Other mutant and transgenic strains were: *Ir25a2*[14], *Ir76b*[27], *Ir25a-Gal4*[25], *Ir76b-Gal4* (insertions on chromosome 2 or 3)[20], *Gr5a-Gal4*[58], *Gr64f-LexA*[52], *Gr66a-LexA*[59], *Gr66a-Gal4*[48], *NompC-LexA*[44], *E409-Gal4*[9], *UAS-Ir25a*[25], *UAS-Ir76b*[7], *UAS-GCaMP3*[60], *UAS-mCD8:GFP*[61], *UAS-CD4:tdTomato*[62], *UAS-mCD8:RFP*[63], *LexAop-mCD8:GFP-2A-mCD8:GFP*[56], *LexAop-rCD2:GFP*[56], *UAS-CsChrimson*[64].

**CRISPR/Cas9-based genome editing**. *Ir56d1* and *Ir56d2*: we identified two CRISPR target sequences within the *Ir56d* locus using ZiFiT (zifit.partners.org/ZiFiT)[65] that are both unique within the genome and which contain an adjacent 3' protospacer adjacent motif (PAM) (Fig. 6a). We generated DNA templates for synthetic guide RNA synthesis by PCR using standard procedures[66] using the following oligonucleotides: CRISPRsgR with either CRISPRsgf-*Ir56d1* or CRISPRsgf-*Ir56d2* (Supplementary Table 3). The template was transcribed in vitro with T7 polymerase, RNA was microinjected into *vas-Cas9* flies (expressing Cas9 specifically in the germline)[67] and mutations in the target sequence region screened by Genetic Services, Inc. After establishment of homozygous mutant lines, mutations were reconfirmed by Sanger sequencing.

*Ir56dGal4*: the *Gal4* knock-in allele was generated via CRISPR/Cas9 mediated homologous recombination. Two sgRNAs targeting the *Ir56d* locus were cloned into pCFD5[68] by Gibson Assembly to generate *pCFD5-Ir56dsgRNAs*. Homology arms for the *Ir56d* locus were fused to the Gal4-hsp70-3'UTR by PCR amplification using genomic DNA and *pGal4attB*[15] as templates, respectively. The product was ligated into *pHD-DsRed-attP*[67] after digestion with *SapI* and *AarI* (to generate the donor vector *pHD-Ir56d-Gal4-DsRed-attP*). *pCFD5-Ir56dsgRNAs* and *pHD-Ir56d-Gal4-DsRed-attP* were co-injected into *Act5C-Cas9,lig4[169]* flies[69] following standard protocols. Successful integration events were identified by screening for DsRed expression and diagnostic PCR. Subsequently, the *DsRed* marker was removed by injection of Cre recombinase. The oligonucleotides used are listed in Supplementary Table 3 and Supplementary Fig. 3a depicts a schematic of the *Ir56dGal4* allele before and after *DsRed* removal.

**Histology**. Immunofluorescence on peripheral and central tissues from larvae and adult flies was performed following standard procedures[28,45]. Primary antibodies:

rabbit anti-IR25a (1:500)[14], guinea pig anti-IR25a (1:200)[21], mouse anti-GFP (1:500; Invitrogen), chicken anti-GFP (1:500; Abcam), rabbit anti-RFP (1:500; Abcam) and mouse monoclonal nc82 (1:10; Developmental Studies Hybridoma Bank). Secondary antibodies (all diluted 1:100–200): goat anti-mouse Alexa 488 (Invitrogen), goat anti-rabbit Cy3 (Milan Analytica, AG), goat anti-chicken Alexa488 (Abcam), goat anti-guinea pig Cy5 (Abcam) and goat anti-mouse Cy5 (Jackson ImmunoResearch). Images were collected with a Zeiss LSM 710 inverted laser scanning confocal microscope (Zeiss, Oberkochen, Germany), and processed with ImageJ and Fiji.

**Optical imaging**. Subesophageal zone imaging: Imaging was performed adapting previous protocols[70,71]. In brief, a 1–3 week-old fly was cold-anaesthetised and inserted into a plastic holder glued to a custom Plexiglas chamber. The head and proboscis of the animal were separated by a plastic barrier that prevents contact between the buffer solution applied to the brain, and the tastant solution. The proboscis was extended using a blunted syringe needle (30 g Blunt, Warner Instruments #SN-30) connected to a vacuum pump (KNF Laboport #N86KN.18) and kept extended using UV curing glue (Tetric EvoFlow, A1, Ivoclar Vivadent) solidified using a UV lamp (Bluphase C8, Ivoclar vivadent). Heads were fixed using the same UV glue and covered with Adult Haemolymph like-Saline buffer (in mM: 108 NaCl, 5 KCl, 2 CaCl₂, 8.2 MgCl₂, 4 NaHCO₃, 1 NaH₂PO₄, 15 Ribose, 5 HEPES; pH 7.5; 265 mOsm). Brains were exposed by removing the cuticle using a microsurgical knife (Sharpoint, Surgical Specialties #72-1501). Complete exposure of the subesophageal zone required the removal of the oesophagus. Delivery of the tastants was performed manually upon the emission of an acoustic signal at frame 20 after the onset of the recording, using a blunted 30 g syringe needle place on a 1 ml syringe containing the solution (BD Plastipak #300013) and mounted on a micromanipulator (Narishige).

Images were acquired with a CCD camera (CoolSNAP-HQ2 Digital CameraSystem) mounted on a fluorescence microscope (upright fixed stage Carl Zeiss Axio Examiner D1) equipped with a 40x water-immersion objective (W Plan-Apochromat 40× /1,0 VIS-IR DIC). Excitation light of 470 nm was produced with an LED light (Cool LED pE-100, VisiChrome). Binned image size was 1000 × 700 pixels on the chip, corresponding to 250 × 175 μm in the preparation. Exposure time was 100 ms. Twenty-second films were recorded with an acquisition rate of 4 Hz. Metafluor software (Visitron) was used to control the camera, light, and data acquisition.

Data were processed using NIH ImageJ and custom programmes in Matlab (v9.0). Time-series images corresponding to one experiment were first aligned using StackReg/TurboReg (big www.epfl.ch/thevenaz/stackreg/) in NIH ImageJ. Raw images were then segmented into individual 80-frame measurements. Each measurement was bleach-corrected by fitting a double-exponential function to the relative mean fluorescence in the ROI over time, excluding the frames covering 12.5 s after stimulus onset. We then calculated the relative change in fluorescence ($\Delta F/F$) for each frame of each measurement as $(\Delta F_i = F_0)/F_0 \times 100$, where $F_0$ is the mean fluorescence value of frames 10–15 (before tastant presentation at frame 20), and $F_i$ is the fluorescence value for the $i$th frame of the measurement. A circular ROI (diameter 7 pixels) was used for quantification of all measurements from the same animal. The maximal $\Delta F/F$ between frames 20 and 60 for each stimulus was used for data representation and statistical analysis.

Foreleg calcium imaging: imaging was performed adapting previous protocols[33]. A custom-made bottom-glass imaging chamber was built by drilling a 10 mm hole in a 35 mm Petri dish (Falcon #351008) onto which an 18 × 18 mm coverslip (Menzel-Gläser #631-1331) was glued. A 1–3 week-old fly was cold-anaesthetised and the forelegs cut with a razor blade between the femur and the tibia. UV-curing glue was used to seal the cut end of the leg, which was then mounted laterally on the glass surface of the imaging chamber. 1% low melting point agarose (Peqlab #35–2010) was used to cover the leg leaving exposed only the fourth and fifth tarsal segments. The preparation was covered with 150 μl of milliQ filtered water. For the stimulations, 150 μl of the desired chemical were added manually to the preparation. The sample was subsequently washed five times using milliQ filtered water and left with 150 μl of water for equilibration during 3 min before the next stimulation. Imaging was performed using an inverted confocal microscope (Zeiss LSM 710) using an oil immersion 40x objective (Plan Neofluar 40x Oil immersion DIC objective with a 1.3 NA). 40 frames were taken per stimulation at 250 ms/frame. Different focal planes were used to image the 5b and 5s sensilla neurons. Images were analysed as indicated in the previous section.

**Behaviour**. Two-choice positional preference assay: assays were performed in 94 mm Petri dishes (Greiner-bio-one #632180; 94 × 16 mm), divided into two halves (A and B) by placing two stacks of three-layered semicircles of blotting paper (Macherey-Nagel #742113) separated by a 3–5 mm gap. Prior to the start of the experiment each semicircle stack was soaked with 3 ml of the desired test solution (see below and Supplementary Table 2). Up to 16 arenas were placed on a methacrylate panel (1.5 cm thickness) elevated 5.5 cm from the light source, which consisted of a 60 × 60 cm LED Panel (Ultraslim LED Panel, 360 Nichia LEDs, Lumitronix) covered with red film (106 Primary Red, Showtec). 70–80 flies (mixed sexes; 2–3 days old, starved for 24 h in glass culture tubes with a Kimwipe (Kimtech #7552) soaked with 2 ml of tap water; cold anaesthetised) were introduced into the centre of each arena and the lids replaced. When all flies had recovered mobility,

the assay was started. Pictures were taken (using a USB 3.0 100 CMOS Monochrome Camera 2048 × 2048 Pixel and a CCTV Lens for 2/300f:16 mm (iDS)) every 10 min up to 90 min using a custom Matlab code. The distribution of animals in the arena at 90 min (excluding the rare flies that were non-motile or that died during the assay) was quantified using a custom macro in Fiji (code available upon request). Preference indices were calculated as: (# flies in A − # flies in B)/total # flies. For the experiments in Fig. 7b, different genotypes were run in parallel, and randomised with respect to arena position.

For carbonation preference tests, in order to ensure a slow but constant production of $CO_2$ over the course of the assay, we used solutions of freshly-prepared 100 mM $NaHCO_3$ that were adjusted to pH 6.5 (with 5 M $NaH_2PO_4$; ~1–1.5 ml/100 ml) for the carbonated side and pH 8.5 (with NaOH; <50 μl/100 ml) for the non-carbonated side[9]. To test for preference due to pH, we use phosphate buffered saline (7.8 mM $NaH_2PO_4$, 12.2 mM $Na_2HPO_4$, 153.8 mM NaCl) solutions at pH 6.5 or 8.5 (Fig. 7c). To eliminate the possibility that preference differences were due to $Na^+$ imbalance (due to a larger volume of 5 M $NaH_2PO_4$ required to set the pH of $NaHCO_3$ at pH 6.5 than NaOH to set the pH to 8.5), we supplemented the $NaHCO_3$ pH 8.5 solution with NaCl to achieve an ~150 mM $[Na^+]$ in both test solutions; flies retained the preference for the carbonated solution (Fig. 7d).

Proboscis extension reflex (PER) assay: PER in response to labellar stimulation was assessed following a standard protocol[72]. Individual flies (mixed sexes; 3–5 days old, starved for 24 h) were introduced into yellow pipette tips (Starlabs #S1111.0706), whose narrow end was cut in order that only the fly's head could protrude from the opening, leaving the rest of the body, including legs, constrained inside the tip. Tastants (Supplementary Table 2) were delivered using a piece of Kimwipe. Each fly was first tested with water; where this caused PER, water was offered *ad libitum*, and the animal tested again. Only flies that showed negative PER for water were assayed with the other stimuli. Up to six flies were prepared simultaneously and tastants were randomised across trials.

For leg stimulation-evoked PER we adapted a published protocol[33]. Groups of six cold anesthetized flies (mixed sexes; 3–5 days old, starved for 24 h) were glued on their back on top of 76 × 26 mm microscope slides (Menzel-Gläser; #631–0649) using UV curing glue (Tetric EvoFlow, A1, Ivoclar Vivadent) solidified using a UV lamp (Bluphase C8, Ivoclar vivadent). Up to five groups of six flies were prepared at the same time and allowed to recover for 30 min in a humidified chamber. Only one group of six flies was tested at a time for the whole set of stimuli presented using a piece of Kimwipe[72]. Before the beginning of the stimulations, flies were allowed to drink water ad libitum. Each stimulus was presented once allowing the flies to touch the Kimwipe with all the legs for 5 s. Only full extensions of the proboscis were considered as positive responses. Between stimuli, flies were allowed to drink water ad libitum and the legs washed with water.

Optogenetic induction of PER: flies were grown on standard food; prior to the experiment 3–5 days old flies were starved for 24 h in tubes containing a Kimwipe soaked in 2 mM all-*trans*-retinal (Sigma #R2500) in tap water. Flies were cold-anaesthetised and glued on their backs to the external side of a 94 × 16 mm plastic plate using UV curing glue (see above). Groups of 6–8 flies of the same genotype were prepared in a row and tested for PER to stimulation by a 650 nm laser diode (1 mW, Adafruit Industries #1054) aimed at the proboscis with an intensity of 2–2.5 μW/mm². Only full proboscis extensions were considered as positive.

Expresso food ingestion measurements and analysis: flies were maintained on conventional cornmeal-agar-sucrose medium at 23–25 °C and 60–70% relative humidity (RH), under a 12 h light:12 h dark cycle (lights on at 6 am). Carbonated and non-carbonated control solutions were prepared as described above (either in water or with 5 mM sucrose). Food ingestion was measured in the Expresso device:[47] individual flies (2–5 days old male $w^{1118}$ flies, starved 24 h) were placed in the behavioural chamber with the doors in the closed position to prevent access to the liquid food in the calibrated glass capillaries. Expresso data acquisition software was started at which point all doors were opened giving flies synchronised access to liquid food. Each trial lasted ~33 min, and 10 flies were tested in parallel in two Expresso sensor banks. For each condition, 20–30 flies were tested. The measurements were performed at Zeitgeber Time 6–10. The Expresso food ingestion data were analysed using a custom programme in Python (available upon request). The change points in the Expresso signal that denote a meal bout and the amount of food ingested were detected using the Pruned Exact Linear Time algorithm. Total ingestion was calculated as the total volume ingested per fly per trial. The latency was calculated as the time before the first meal after door opening. When a fly did not consume any food, the total meal bout volume was scored as 0 and latency to first meal bout was scored as the total time of the assay (i.e. 33 min). All data were analysed in R statistical software.

flyPAD: we assayed mated $w^{1118}$ female flies that were reared at 25 °C, 70% RH on a 12 h light:12 h dark cycle. Flies were reared at standard density and were matched for age and husbandry conditions. The fly medium contained, per litre, 80 g cane molasses, 22 g sugar beet syrup, 8 g agar, 80 g corn flour, 10 g soya flour, 18 g yeast extract, 8 ml propionic acid, and 12 ml nipagin (15% in ethanol). The day before the assay the fully fed flies were flipped into new vials to ensure a fully fed metabolic state. The starvation state was induced by transferring flies for 24 h before the assay into vials containing a tissue soaked with water.

flyPAD assays[53,54] were performed using only one well of the arena per assay. The well was filled with a solid substrate comprising 20 mM sucrose made with either non-carbonated or carbonated water (Supplementary Table 2) in 1% agarose.

Flies were individually transferred to flyPAD arenas by mouth aspiration and allowed to feed at 25 °C and 70% RH for 60 min. flyPAD data were acquired using the Bonsai framework[73], and analysed in MATLAB using custom-written software[54].

**Statistics.** Sample size was determined based upon preliminary experiments. Data were analysed and plotted using R (v1.0.153; R Foundation for Statistical Computing, Vienna, Austria, 2005; R-project-org) (code available upon request). Except for PER and flyPAD experiments, quantitative data are represented showing their distribution by superimposing a boxplot on top of a violin plot. The violin plot shows the kernel density estimate; for the boxplots the whiskers are calculated as follows: the upper whisker equals the third quartile plus 1.5× the interquartile range (IQR) and the lower whisker equals the first quartile minus 1.5× the IQR. Any data points above the superior or below the inferior whisker values are considered as outliers. The outliers were included in the statistical comparisons as we performed non-parametric rank tests. Data were analysed statistically using different variants of the Wilcoxon test, except where indicated. For comparisons between distributions, the Wilcoxon rank sum test was used. When P value correction for multiple comparisons was required, the Bonferroni method was used. For the experiments in Fig. 7b–d, we performed a Wilcoxon Signed Rank Test with the null hypothesis that the median of sampled values differs from zero. For PER results we used the Fisher exact test. For Expresso assay data, pairwise comparisons using the Tukey and Kramer (Nemenyi) test with Tukey-Dist approximation for independent samples were performed. flyPAD results were compared using the Wilcoxon rank-sum test.

## Data availability
All relevant data supporting the findings of this study are available from the corresponding author on request.

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

## Acknowledgements
We thank Carolina Gomez-Diaz for assistance with CRISPR primer design, and acknowledge the Bloomington *Drosophila* Stock Centre (NIH P40OD018537) and the Developmental Studies Hybridoma Bank (NICHD of the NIH, University of Iowa) for reagents. We thank members of the Benton lab for discussions and comments on the manuscript. We thank Célia Baltazar for assistance in performing the flyPAD assays. J.A. S.-A. was supported by a Federation of European Biochemical Societies Long Term Fellowship, an EMBO Long Term Fellowship and a Human Frontier Science Programme Long-term Fellowship. V.C. was supported by a Boehringer Ingelheim Foundation Fellowship. T.O.A. was supported by a Human Frontier Science Programme Long-term Fellowship. D.M. was supported by a DFG Research Fellowship (MU 4116/1-1). Research in S.G.S.'s laboratory was supported by a European Research Council Starting Independent Research Grant (309832) and the Swiss National Science Foundation (31003A_149499). Research in N.Y.'s laboratory was supported by a Cornell University Nancy and Peter Meinig Family Investigatorship Programme, a Pew Biomedical Scholar Award, and the Alfred P. Sloan Foundation Award. Research in C.R.'s laboratory was supported by the BIAL Foundation grant (279/16) and the Champalimaud Foundation. Research in R.B.'s laboratory was supported by the University of Lausanne, ERC Starting Independent Researcher and Consolidator Grants (205202 and 615094) and an SNSF Sinergia Grant (CRSII3_136307).

## Author contributions

The authors have made the following declarations about their contributions: J.A.S.-A., A.F.S., V.C., A.K.S., S.Y.S., D.M., K.S., T.O.A., G.L.N.-M., S.G.S., N.Y., C.R., and R.B. conceived and designed the experiments. J.A.S.-A. (Fig. 4, Figs 6 and 7; Supplementary Fig. 3, 4, 6 and 7), A.F.S. (Fig. 5), V.C. (Figs 1 and 2; Supplementary Fig. 1), G.Z. (Fig. 7a–d), A.K.S. (Fig. 1, Fig. 3, Supplementary Fig. 2), S.Y.S. (Supplementary Fig. 5), T.O.A. (Supplementary Fig. 3), S.C. (Fig. 3), G.L.N.-M. (Fig. 2), R.B. (Fig. 6a), K.S. and D.M. (Supplementary Fig. 8) performed the experiments. J.A.S.-A., A.F.S., V.C., A.K.S., S.Y.S., D.M., K.S., T.O.A., G.L.N.-M., N.Y., C.R. and R.B. analysed the data. J.A.S.-A., A.F.S., V.C., L.A., T.O.A. and R.B. contributed reagents/materials/analysis tools. J.A.S.-A. and R.B. wrote the paper, with input from all authors.

## Additional information

**Competing interests:** The authors declare no competing interests.

