## [Peer Review File · Nature Communications]

Reviewers' comments:

Reviewer #1 (Remarks to the Author):

The manuscript from Benton and colleagues provides a rich analysis of the *Drosophila*, chemosensory Ionotropic Receptors (IRs). First they provide a thorough anatomical description of the complete collection of IR expressing sensory neurons, and generate a large toolbox of Gal4 lines that will be useful to *Drosophila* researches. The authors then focus on a particular IR, Ir56d, and show that the sensory neurons expressing this receptor in the labellum respond to carbonation in an Ir56d-dependent manner. Thus, Ir56d is the receptor that is responsible for *Drosophila* detection of carbonation. Moreover, this receptor is required for attraction to carbonation in a two-choice place preference assay. Overall, this is an excellent study, and the experiments are well-conceived and elegant. I have only minor comments.

1. The use of statistics throughout is solid but I am confused about the "outliers" in the violin plots throughout the paper. The legend in Figure 5 states, "Black dots represent outliers." Are these statistically determined to be outliers? And most importantly, are they excluded in the statistical comparisons between groups? The authors should clarify this in the methods.

2. IR57d is expressed in the labellum as well as the legs. The authors perform calcium-imaging experiments that demonstrate that the labellum IR57d neurons respond to carbonation. However, stimulation of these neurons does not drive appetitive behavior in the PER assay. Is the attraction to carbonation observed in the two-choice assay is due to those in the legs? The authors have not shown that these neurons also respond to carbonation, but it is inferred if Ir56d is the receptor. The authors don't need to sort this out, but they should acknowledge this gap in logic.

Reviewer #2 (Remarks to the Author):

Sanchez-Alcaniz and colleagues describe the expression of IR genes expressed in the taste system, and they provide a more detailed analysis of one of these IR genes, IR56d, in sensing carbonation. The expression analysis of the IR genes is well done, comprehensive and provides a useful tool for researchers in the field.

The analysis of IR56d gene and the more broadly expressed IR76b and IR25a genes in carbonation is interesting, identifying set of IR proteins that clearly are part of a receptor for this stimulus. This is novel, but when put in perspective, it represents a minor advance overall. First, the neurons mediating carbonation taste have long been identified (Fischler et al., 2007), but that taste quality is at best of curious relevance to any feeding related behavior (unless one considers "staying on a substrate without eating it" interesting). Thus, it would be unquestionably enlightening if there would be a behavior that does make some sense to feeding etc. In addition, there are numerous issues related to the cellular and molecular basis of carbonation that the authors should test and discuss (see below). Lastly, the authors make unfounded and exaggerated claims about their study with regard to a "controversy", the role of a Gr gene in fatty acid taste. Indeed, the data presented on fatty acid taste are out of place and provide no new information than what already has been published.

Overall, the work on the expression is useful, but not interesting to most readers (except insect researchers working on taste). It is disconnected from the second part of the story (carbonation). The identification of IRs involved in carbonation is interesting, but incomplete, while the claim of "extension and clarification of contradictory data" on IRs and Grs is simply not accurately reflecting

what these studies actually report. Thus, unless the authors identify a relevant behavior for carbonation taste and address the points below, the paper is not suitable for Nature Communication.

Major points:

The two parts of the paper are disconnected. Especially the first part should be published on its own, in a specialty journal intended for fly/insect taste investigators. While very valuable and well-done, and of value to a group of investigators, it is rather irrelevant for the general audience. This part of the work is more suitable for a specialty journal.

As for the role of IR56d in carbonation and fatty acid taste, the presented data is incremental and/or a confirmation of previous studies. The only new finding is that IRs mediate carbonation, yet that aspect of the study is rather incomplete. A main question, what the biological relevance of the observed preference for carbonation is, is not addressed. Since carbonation does not lead to PER, does not enhance feeding etc., the reader is at a loss of what do think of this obscure taste modality.

In addition, the authors should address is the role of tarsal neurons expressing these three receptors (Tauber et al, Ahn et al., 2017). Do these neurons respond to carbonation? If so, do they contribute to the preference? Perhaps PER requires activation of both peg and tarsal neurons, or perhaps carbonation enhances PER to fatty acid, but not sugars. These are simple experiments that could shed light on the possibility that carbonation is indeed somehow a modulatory cue for feeding.

The authors give the impression of a controversy with regard IRs in fatty acid taste, referring to a role of a sugar Gr gene in this taste modality (Kim et al., 2018). However, and as they also mention, Kim et al.'s data rule out that Or64e is the receptor itself, but acts downstream of it. Furthermore, the "contradictory result" they refer is not contradictory at all, albeit perhaps a bit surprising, dealing with Gr64e's role in PER. In contrast to Kim et al., Ahn et al. reported that PER to fatty acids is not affected in flies lacking all sugar Gr genes (including Gr64e). However, the two groups use different experimental set ups: In Kim et al., the labellum is stimulated, while in Ahn et al, the tarsi are stimulated. Thus, there might simply be different signaling pathways for fatty acid taste in these organs. This is already evident from the Ahn et al. study, which also showed that bitter neurons responding to fatty acid do so independently of IR proteins. Thus, the statement at the end of the introduction ("Furthermore, we extend and clarify recent, partially conflicting, studies...") is not only wrong but also does not address the issue at hand at all. And where their supposed clarification comes to play is not clear to this reviewer.

Overall, the data on fatty acid taste do not add any new information that was not already published by Tauber et al. (2017) and Ahn et al (2017). It is fine to mention these experiments, but they should not be in the main section of the paper.

Finally, the authors should address a curious observation, which must have escaped their attention: LeDue et al. (2015), which is not cited in the paper, reported that taste peg neurons expressing Gr64e-GAL4 respond to carbonation, and hence the authors should investigate (i) whether this driver is co-expressed with Ir56d and (ii), if so, whether Gr64e is required for such responses.

Minor points:

P 3: Pan-repertoire? Odd expression,

Same sentence: " ..., which we use to survey of the expression..." delete the word "of"

P 5: statement "IR56d-Gal4 is the only reporter expressed in neurons housed in the taste pegs, ..." is obviously incorrect, as both IR25a-Gal4 and IR76b-GAL4 are also expressed in these neurons.

P 6: The use of Gr5a as a sugar neuron marker is inadequate. It has been reported multiple times that

transgenes as well as gene knock-ins into this locus are expressed more broadly and NOT restricted to sweet neurons in the labellum (Fujii et al. 2015). The authors should use Gr64f-GAL4 as a marker for sugar sensing neurons.

P 7: seems misrepresentation to say that the “final position” was used to calculate the preference index. They probably recorded position every certain interval during the 90 minutes to calculate the preference index. Clarify.

P 9: the in vivo expression (first line in discussion). As opposed to? “in vivo” is not necessary.

P10: The statement “The taste bristle population of Ir56d neurons represents a subset of sugar-sensing neurons...” is not supported by rigorous analysis, because of the inadequate Gal4 driver used (Gr5a).

Reviewer #3 (Remarks to the Author):

This is an important study that characterizes the expression of a large number of IRs in the gustatory system and determines that three IRs are expressed in carbonation-sensing neurons and are required for the CO₂ response. The study is very exciting and the data are of high quality. I think that this is appropriate for publication with only minor changes.

1) One question is how closely the IR56d cells correspond to the CO₂-sensing cells marked by E409. Figure 4E is not very clear—seems to show labellar Ir56d more than taste pegs. Focusing on taste pegs in a whole mount similar to 4A or switching red and green reporters or showing overlap in SEZ projections would be useful to determine this.

2) The identification of IRs as candidate CO₂ sensors is important and provides insight that extends well beyond previous studies characterizing CO₂-sensing cells in the E409 line. Yet, as all experiments are consistent with the previous work, some sentences seem to be unnecessarily dismissive. I would recommend minor sentence changes.

-Change sentence p. 6 to “These data – together with our co-expression analysis (Fig. 4e) – identify the Ir56d taste peg neurons as the carbonation-sensing cells that were previously identified by their expression of the E409-Gal4 enhancer trap9.”

-Change sentence p. 7 to “However, the requirement of E409-Gal4 taste cells in this response was difficult to determine because the E409-Gal4 enhancer trap available at the time of that study is expressed in many central neurons in addition to the taste pegs9, limiting its usefulness for neuronal manipulation experiments.”

-Change sentence p. 7 to “These observations are consistent with those made using a different positional-preference assay9, confirming that carbonation (a product of microbial fermentation) is a modestly attractive stimulus for Drosophila.”

Reviewer 1

The manuscript from Benton and colleagues provides a rich analysis of the *Drosophila*, chemosensory Ionotropic Receptors (IRs). First they provide a thorough anatomical description of the complete collection of IR expressing sensory neurons, and generate a large toolbox of Gal4 lines that will be useful to *Drosophila* researchers. The authors then focus on a particular IR, Ir56d, and show that the sensory neurons expressing this receptor in the labellum respond to carbonation in an Ir56d-dependent manner. Thus, Ir56d is the receptor that is responsible for *Drosophila* detection of carbonation. Moreover, this receptor is required for attraction to carbonation in a two-choice place preference assay. Overall, this is an excellent study, and the experiments are well-conceived and elegant. I have only minor comments.

1. The use of statistics throughout is solid but I am confused about the “outliers” in the violin plots throughout the paper. The legend in Figure 5 states, “Black dots represent outliers.” Are these statistically determined to be outliers? And most importantly, are they excluded in the statistical comparisons between groups? The authors should clarify this in the methods.

RESPONSE: We present violin plots with boxplots superimposed, in which the whiskers are calculated as follows: the upper whisker equals the third quartile plus 1.5x the Interquartile range (IQR) and the lower whisker equals the first quartile minus 1.5x the IQR. Any data points above the superior or below the inferior whisker values are considered as outliers. The outliers are included in the statistical comparisons as we are performing a statistical non-parametric rank test. We have modified the Methods section on Statistics to include this information.

2. IR57d [IR56d] is expressed in the labellum as well as the legs. The authors perform calcium-imaging experiments that demonstrate that the labellum IR57d [IR56d] neurons respond to carbonation. However, stimulation of these neurons does not drive appetitive behavior in the PER assay. Is the attraction to carbonation observed in the two-choice assay is due to those in the legs? The authors have not shown that these neurons also respond to carbonation, but it is inferred if Ir56d is the receptor. The authors don't need to sort this out, but they should acknowledge this gap in logic.

RESPONSE: We have performed additional experiments examining the expression of the *Ir56d-Gal4* reporter in tarsal neurons, the physiological sensitivity of these cells to carbonation, as well the ability of carbonation stimulation of tarsi to evoke the proboscis extension reflex (PER). The results, reported in Supplementary Fig. 4, indicate that *Ir56d* is only expressed in sugar-sensing neurons in the tarsi (labelled by a *Gr64f* reporter) – which is consistent with previous observations (Ahn et al., eLife 2017) – and that carbonation does

not evoke detectable calcium responses in these cells. Consistently, selective carbonation stimulation of legs does not evoke PER. These observations indicate that tarsal *Ir56d*-expressing neurons are molecularly and functionally similar to the labellar taste bristle population of *Ir56d*-positive neurons (i.e., sugar- but not carbonation-sensitive). Thus, the labellar taste peg neurons are likely to be the principal (and potentially only) carbonation sensors underlying positional preference in the two-choice assay.

Reviewer #2

Sanchez-Alcaniz and colleagues describe the expression of IR genes expressed in the taste system, and they provide a more detailed analysis of one of these IR genes, IR56d, in sensing carbonation. The expression analysis of the IR genes is well done, comprehensive and provides a useful tool for researchers in the field. The analysis of IR56d gene and the more broadly expressed IR76b and IR25a genes in carbonation is interesting, identifying set of IR proteins that clearly are part of a receptor for this stimulus. This is novel, but when put in perspective, it represents a minor advance overall. First, the neurons mediating carbonation taste have long been identified (Fischler et al., 2007), but that taste quality is at best of curious relevance to any feeding related behavior (unless one considers “staying on a substrate without eating it” interesting). Thus, it would be unquestionably enlightening if there would be a behavior that does make some sense to feeding etc. In addition, there are numerous issues related to the cellular and molecular basis of carbonation that the authors should test and discuss (see below). Lastly, the authors make unfounded and exaggerated claims about their study with regard to a “controversy”, the role of a Gr gene in fatty acid taste. Indeed, the data presented on fatty acid taste are out of place and provide no new information than what already has been published.

Overall, the work on the expression is useful, but not interesting to most readers (except insect researchers working on taste). It is disconnected from the second part of the story (carbonation). The identification of IRs involved in carbonation is interesting, but incomplete, while the claim of “extension and clarification of contradictory data” on IRs and Grs is simply not accurately reflecting what these studies actually report. Thus, unless the authors identify a relevant behavior for carbonation taste and address the points below, the paper is not suitable for Nature Communication.

Major points:

The two parts of the paper are disconnected. Especially the first part should be published on its own, in a specialty journal intended for fly/insect taste investigators. While very valuable and well-done, and of value to a group of investigators, it is rather irrelevant for the general audience. This part of the work is more suitable for a specialty journal. As for the role of IR56d in carbonation and fatty acid taste, the presented data is incremental and/or a confirmation of previous studies. The only new finding is that IRs mediate carbonation, yet that

aspect of the study is rather incomplete. A main question, what the biological relevance of the observed preference for carbonation is, is not addressed. Since carbonation does not lead to PER, does not enhance feeding etc., the reader is at a loss of what do think of this obscure taste modality.

RESPONSE: We submitted our work to this journal because of its aim “to represent important advances of significance to specialists within each field” (from Nature Communications, Aims and Scope). Moreover, the field of insect taste is not inconsequential, and widely recognised for providing general insights into understanding how sensory stimuli are recognised and encoded to control behaviour. It is also possible that some of the non-antennal IRs whose expression we characterise in this work will function in other sensory modalities (just as the antennal-expressed IRs mediate olfaction, thermosensation and hygrosensation).

The original identification of carbonation as a taste modality in *Drosophila* is an important paper in the field (Fischler et al., Nature 2007), and we believe that the identification of IRs as at least part of the underlying molecular mechanisms of sensory detection is a non-incremental result. We confirm the original observations of the earlier work in showing that carbonation evokes behavioural attraction in a positional preference assay, and extend them by showing that this depends upon IR56d. However we acknowledge that many questions regarding the precise behavioural function of this sensory stimulus remain open. We should stress this is not through lack of effort: we have made substantial investment into this problem over several years, using a variety of behavioural paradigms. Rather, it appears that regulation of feeding behaviours in *Drosophila* is a complex process, where not all individual stimuli (such as carbonation) will necessarily give rise to easily quantifiable behaviours (such as PER). Nevertheless, we believe that publication of our work should inspire further efforts in the field.

In addition, the authors should address is the role of tarsal neurons expressing these three receptors (Tauber et al, Ahn et al., 2017). Do these neurons respond to carbonation? If so, do they contribute to the preference? Perhaps PER requires activation of both peg and tarsal neurons, or perhaps carbonation enhances PER to fatty acid, but not sugars. These are simple experiments that could shed light on the possibility that carbonation is indeed somehow a modulatory cue for feeding.

RESPONSE: We have performed additional experiments examining the expression of the *Ir56d-Gal4* reporter in tarsal neurons, the physiological sensitivity of these cells to carbonation, as well the ability of carbonation stimulation of tarsi to evoke the proboscis extension reflex (PER). The results, reported in Supplementary Fig. 4, indicate that *Ir56d* is only expressed in sugar-sensing neurons in the tarsi (labelled by a *Gr64f* reporter) – which is consistent with previous observations (Ahn et al., eLife 2017) – and that carbonation does not evoke detectable calcium responses in these cells. Consistently, selective

carbonation stimulation of legs does not evoke PER. These observations indicate that tarsal *Ir56d*-expressing neurons are molecularly and functionally similar to the labellar taste bristle population of *Ir56d*-positive neurons (i.e., sugar- but not carbonation-sensitive). Thus, the labellar taste peg neurons are likely to be the principal (and potentially only) carbonation sensors underlying positional preference in the two-choice assay.

The authors give the impression of a controversy with regard IRs in fatty acid taste, referring to a role of a sugar Gr gene in this taste modality (Kim et al., 2018). However, and as they also mention, Kim et al.'s data rule out that Or64e [Gr64e] is the receptor itself, but acts downstream of it. Furthermore, the “contradictory result” they refer is not contradictory at all, albeit perhaps a bit surprising, dealing with Gr64e's role in PER. In contrast to Kim et al., Ahn et al. reported that PER to fatty acids is not affected in flies lacking all sugar Gr genes (including Gr64e). However, the two groups use different experimental set ups: In Kim et al., the labellum is stimulated, while in Ahn et al, the tarsi are stimulated. Thus, there might simply be different signaling pathways for fatty acid taste in these organs. This is already evident from the Ahn et al. study, which also showed that bitter neurons responding to fatty acid do so independently of IR proteins. Thus, the statement at the end of the introduction (“Furthermore, we extend and clarify recent, partially conflicting, studies...”) is not only wrong but also does not address the issue at hand at all. And where their supposed clarification comes to play is not clear to this reviewer. Overall, the data on fatty acid taste do not add any new information that was not already published by Tauber et al. (2017) and Ahn et al (2017). It is fine to mention these experiments, but they should not be in the main section of the paper.

RESPONSE: We apologise for the misleading wording and have rephrased the presentation and discussion of our and others' data on fatty acid detection in a more straightforward manner. In particular, we have taken care to highlight that the different studies uncovered apparently distinct fatty acid sensing mechanisms in different sensory appendages (i.e., GR-dependent or -independent in the labellum and the legs, respectively).

Only one of the previous studies (Ahn et al., eLife 2017) examined the role of IR56d in fatty acid sensing (and only in the legs), using a single transgenic RNAi construct (typically, two or more lines are preferred to avoid issues of potential off-target effects). We have focussed on the role of IR56d in fatty acid sensing in the labellum, and believe that the characterisation of the physiological and behavioural phenotype of our *Ir56d* loss-of-function mutant provides important new results. Nevertheless, as the main focus of our manuscript is on carbonation sensing, we deliberately report the analysis of hexanoic acid sensing in a supplementary figure (new Supplementary Fig. 6), but describe these results in the main text to avoid an inconveniently-placed “supplementary results”.

Finally, the authors should address a curious observation, which must have escaped their attention: LeDue et al. (2015), which is not cited in the paper,

reported that taste peg neurons expressing Gr64e-GAL4 respond to carbonation, and hence the authors should investigate (i) whether this driver is co-expressed with Ir56d and (ii), if so, whether Gr64e is required for such responses.

RESPONSE: The *Gr64e-Gal4* transgenic driver labels two populations of neurons, one in the taste bristles and another in the taste pegs (Wisotsky et al., Nat Neuro 2011), similar to our *Ir56d-Gal4* transgenic reporter. However, it is unclear whether *Gr64e* is endogenously expressed in taste pegs for two reasons. First, a *Gr64e^{LexA}* knock-in allele (a genetic tool that is considered to best reflect endogenous gene activity, especially in tandem clusters of receptors) was reported to be “occasionally observed” in only 1 or 2 taste peg neurons (Fujii et al., Curr Biol 2015). This contrasts with our *Ir56d^{Gal4}* knock-in allele, which fully recapitulates the *Ir56d-Gal4* transgenic reporter expression (Supplementary Fig. 3). Second, although *Gr64e* is required in taste bristles for glycerol responses, and sufficient to confer glycerol sensitivity on other neurons when mis-expressed (Wisotsky et al., Nat Neuro 2011), the *Gr64e-Gal4* positive taste peg neurons do not respond to glycerol (LeDue et al., Nat Comm 2015; Steck et al., eLife 2018; our data (Fig. 5c)), suggesting it may not be expressed in these neurons (although there could, of course, be other explanations).

While we cannot exclude a contribution of GR64e to carbonation sensing (or a contribution of other Grs expressed in taste peg neurons e.g., Fujii et al., Curr Biol 2015), we feel it would be beyond the scope of the current work to investigate this thoroughly. We highlight in the Discussion that our attempts to reconstitute carbonation responses by mis-expression of IRs have failed, and state that this “*suggests that additional molecules or cellular specialisations are required.*”

Minor points:

P 3: Pan-repertoire? Odd expression,

RESPONSE: We have rephrased this sentence:

“Here we describe a set of transgenic reporters for the entire Ir repertoire, ...”

Same sentence: “ ..., which we use to survey of the expression...” delete the word “of”

RESPONSE: We have corrected this phrasing.

P 5: statement “IR56d-Gal4 is the only reporter expressed in neurons housed in the taste pegs, ...” is obviously incorrect, as both IR25a-Gal4 and IR76b-GAL4 are also expressed in these neurons.

RESPONSE: The original statement was in reference to reporters for the non-antennal Irs (which excludes IR76b and IR25a), but we agree it was ambiguously phrased and have clarified this sentence:

“To determine the gustatory function of one of the non-antennal IRs, we focussed on IR56d, motivated by its unique expression: Ir56d-Gal4 is the only reporter – apart from the broadly-expressed Ir25a-Gal4 and Ir76b-Gal4 – detected in neurons housed in the taste pegs”

P 6: The use of Gr5a as a sugar neuron marker is inadequate. It has been reported multiple times that transgenes as well as gene knock-ins into this locus are expressed more broadly and NOT restricted to sweet neurons in the labellum (Fujii et al. 2015). The authors should use Gr64f-GAL4 as a marker for sugar sensing neurons.

RESPONSE: We thank the reviewer for pointing out this technical issue. We have now repeated the co-localisation experiments of Ir56d with sweet neurons using a transgenic reporter for *Gr64f* instead of *Gr5a* (Fig. 4g-h).

P 7: seems misrepresentation to say that the “final position” was used to calculate the preference index. They probably recorded position every certain interval during the 90 minutes to calculate the preference index. Clarify.

RESPONSE: We did indeed record the position of animals every 10 min during these assays, as this allowed us to detect and eliminate the rare animals displaying no mobility. Because, for simplicity, we report the preference index only for the final position at 90 min (when the behavioural response plateaus) – similar to other feeding preference studies (e.g., Zhang et al., Science 2013) – we have simplified the description of the quantification in the results section.

P 9: the in vivo expression (first line in discussion). As opposed to? “in vivo” is not necessary.

RESPONSE: We have deleted “in vivo”.

P10: The statement “The taste bristle population of Ir56d neurons represents a subset of sugar-sensing neurons...” is not supported by rigorous analysis, because of the inadequate Gal4 driver used (*Gr5a*).

RESPONSE: As described above, we have now repeated the co-localisation experiments of Ir56d with sweet neurons using a transgenic reporter for *Gr64f* instead of *Gr5a* (Fig. 4g-h).

Reviewer #3

This is an important study that characterizes the expression of a large number of IRs in the gustatory system and determines that three IRs are expressed in carbonation-sensing neurons and are required for the CO₂ response. The study is very exciting and the data are of high quality. I think that this is appropriate for

publication with only minor changes.

1) One question is how closely the Ir56d cells correspond to the CO₂-sensing cells marked by E409. Figure 4E is not very clear—seems to show labellar Ir56d more than taste pegs. Focusing on taste pegs in a whole mount similar to 4A or switching red and green reporters or showing overlap in SEZ projections would be useful to determine this.

RESPONSE: We have provided new proboscis images in Fig. 4e to more clearly highlight co-expression of *Ir56d-LexA* with *E409-Gal4* in taste peg neurons. No *E409-Gal4*-positive neurons co-localise with the *Ir56d-LexA* expressing taste bristle neurons. In our hands, *E409-Gal4* is a rather weak enhancer trap for the taste neurons, and although we could detect *E409-Gal4*-positive neuronal innervations in the AMS1 region in the SEZ (Reviewer Fig. 1), the low signal – and the many other neurons in the central brain this driver labels – made it difficult to unambiguously examine co-expression with *Ir56d-LexA*-positive neuron axonal innervations. Nevertheless, based upon anatomical and physiological properties we are confident that the Ir56d neurons we have characterised correspond to the previously described *E409-Gal4* carbonation sensors.

E409-Gal4>UAS-Tomato

Reviewer Fig. 1. Brain expression and projections of *E409-Gal4* neurons. nc82 neuropil is in blue. White arrowheads highlight the axons innervating the AMS1 region in the SEZ.

2) The identification of IRs as candidate CO₂ sensors is important and provides insight that extends well beyond previous studies characterizing CO₂-sensing cells in the E409 line. Yet, as all experiments are consistent with the previous work, some sentences seem to be unnecessarily dismissive. I would recommend minor sentence changes.

-Change sentence p. 6 to “These data – together with our co-expression analysis (Fig. 4e) – identify the Ir56d taste peg neurons as the carbonation-sensing cells that were previously identified by their expression of the E409-Gal4 enhancer trap⁹.”

-Change sentence p. 7 to “However, the requirement of E409-Gal4 taste cells in this response was difficult to determine because the E409-Gal4 enhancer trap available at the time of that study is expressed in many central neurons in addition to the taste pegs⁹, limiting its usefulness for neuronal manipulation experiments.”

-Change sentence p. 7 to “These observations are consistent with those made using a different positional-preference assay⁹, confirming that carbonation (a product of microbial fermentation) is a modestly attractive stimulus for

Drosophila.”

RESPONSE: Our intention was certainly not to be dismissive; the Fischler et al. Nature 2007 paper is an important contribution to the field that has been very influential in guiding our work. We have rephrased the three sentences as proposed (with one minor edit in the second sentence), which we agree achieve a better tone.

REVIEWERS' COMMENTS:

Reviewer #1 (Remarks to the Author):

The authors have satisfied my concerns and the manuscript is appropriate for publication in Nature Communications.

Reviewer #2 (Remarks to the Author):

The authors have responded well to criticism of the reviewers. The extensive expression analysis is done extremely well and valuable to researchers. Given that the Fly is such an important model system, I do concur that it should not necessarily be restricted to a more specialized journal. The characterization of carbonation sensing neurons and IR proteins is well done and concerns this reviewer had were addressed.

I do have one remaining concern that I think warrants correction. The last paragraph of the introduction summarizes what is currently known about the role of IRs in taste. However, the authors left out studies published last year on acid taste and fatty acid taste, which have been arguably more conclusive than many of work they cite (sugar sensing by pharyngeal taste neurons, amines by taste bristles and amino acids in larvae and flies etc). The paper on fatty acid (Ahn et al., 2017) defined both cells and IR proteins involved in these processes (IR25a, IR76b and IR56d), the specific ligands these IRs detect, and it also characterizes specific behaviors affected when these IR genes are missing. In addition, a paper by Tauber et al (2017), also identifies IR56d expressing sugar neurons responsible for fatty acid taste. The paper on sour taste (Chen and Amrein 2017) mediated by tarsal neurons, requiring IR25a and IR76b and necessary for oviposition in females is not even cited. These studies should be included in this overview in the introduction. These studies, along with the ones that are presented here, further establish IR25a and IR76b as central, ubiquitous components in most IR based taste receptors, a fact that they authors might also want to reiterate in the discussion, and that more specific additional IRs confer specificities to different ligands/cells.

Reviewer #3 (Remarks to the Author):

The authors have satisfied my previous concerns, and this is appropriate for publication with only minor text changes in my view. The additional data that the 3 IRs do not confer carbonation responses in bristles is important and argues that while the genes are necessary, something else is also required. The study identifies IRs required for CO₂ detection, rather than the molecular basis of detection. In light of this, some statements should be modestly changed.

1. Change Title to: An expression atlas of ionotropic glutamate receptors identifies molecules required for carbonation detection, or along these lines.
2. Change abstract, last sentence, to "defines IRs required for carbonation sensing"
3. Lines 365-376, showing that bristles with the same IRs do not respond to CO₂, should go after lines 315, calcium imaging of taste pegs. It is important to be clear that the imaging shows that these receptors are necessary for CO₂ responses, but not sufficient. Placing the calcium imaging data together would make that clearer to your audience.

NCOMMS-18-06889B: RESPONSE TO REVIEWERS

Reviewer #2 (Remarks to the Author):

The authors have responded well to criticism of the reviewers. The extensive expression analysis is done extremely well and valuable to researchers. Given that the Fly is such an important model system, I do concur that it should not necessarily be restricted to a more specialized journal. The characterization of carbonation sensing neurons and IR proteins is well done and concerns this reviewer had were addressed.

I do have one remaining concern that I think warrants correction. The last paragraph of the introduction summarizes what is currently known about the role of IRs in taste. However, the authors left out studies published last year on acid taste and fatty acid taste, which have been arguably more conclusive than many of work they cite (sugar sensing by pharyngeal taste neurons, amines by taste bristles and amino acids in larvae and flies etc). The paper on fatty acid (Ahn et al., 2017) defined both cells and IR proteins involved in these processes (IR25a, IR76b and IR56d), the specific ligands these IRs detect, and it also characterizes specific behaviors affected when these IR genes are missing. In addition, a paper by Tauber et al (2017), also identifies IR56d expressing sugar neurons responsible for fatty acid taste. The paper on sour taste (Chen and Amrein 2017) mediated by tarsal neurons, requiring IR25a and IR76b and necessary for oviposition in females is not even cited.

These studies should be included in this overview in the introduction. These studies, along with the ones that are presented here, further establish IR25a and IR76b as central, ubiquitous components in most IR based taste receptors, a fact that they authors might also want to reiterate in the discussion, and that more specific additional IRs confer specificities to different ligands/cells.

RESPONSE: We have added the suggested references to the Introduction, with the exception of the Tauber et al. 2017 paper, which does not specifically look at the role of IRs in fatty acid sensing (only the *cells* that express IRs), as this would be out of context in this sentence. However, we fully emphasise the Tauber et al. work (and other studies on fatty acid sensing) in the last section of the Results and the Discussion.

Reviewer #3 (Remarks to the Author):

The authors have satisfied my previous concerns, and this is appropriate for publication with only minor text changes in my view. The additional data that the 3 IRs do not confer carbonation responses in bristles is important and argues that while the genes are necessary, something else is also required. The study identifies IRs required for CO₂ detection, rather than the molecular basis of detection. In light of this, some statements should be modestly changed.

1. Change Title to: An expression atlas of ionotropic glutamate receptors identifies molecules required for carbonation detection, or along these lines.

RESPONSE: We appreciate this point, and have modified the title to be “*An expression atlas of variant ionotropic glutamate receptors identifies a molecular basis of carbonation sensing*”. We feel it is important to indicate that these are variant iGluRs (and not canonical synaptic iGluRs), and deliberately wrote in the first submission “a molecular basis” (as opposed to “the molecular basis”) precisely because of this issue that we have not definitively shown that these are the detectors. To further emphasise this point, we have now replaced “detection” with “sensing”. We do feel that our results (notably the calcium imaging data) justifies this title, and that these IRs are excellent candidates for the direct sensory receptors; to our ear, the suggested phrasing was a little vague.

2. Change abstract, last sentence, to "defines IRs required for carbonation sensing".

RESPONSE: We have changed the indicated sentence.

3. Lines 365-376, showing that bristles with the same IRs do not respond to CO₂, should go after lines 315, calcium imaging of taste pegs. It is important to be clear that the imaging shows that these receptors are necessary for CO₂ responses, but not sufficient. Placing the calcium imaging data together would make that clearer to your audience.

RESPONSE: We note that the demonstration that IR56d neurons in labellar taste pegs, but not taste bristles, respond to carbonation already indicated the necessity but not sufficiency of the IRs for carbonation responses. We appreciate the point of this reviewer, and have now moved the calcium imaging data on leg IR56d neurons to the end of the “*IR56d taste peg neurons are gustatory carbonation sensors*” section, where we feel it is more logically placed, given we only characterised leg neuron responses in wild-type animals. We retain the description of the behavioural characterisation of carbonation stimulation of leg tarsi in the original section (with some minor rephrasing for clarity).

Finally, according to the editorial request to avoid citing unpublished data, we have now added a new Supplementary Figure 7, illustrating the failure to reconstitute carbonation responses in heterologous neurons through mis-expression of IR56d, IR25a and IR76b, further emphasising that additional molecular and/or cellular specialisations remain to be discovered for this sensory modality.